# LEARNING HIDDEN CASCADES VIA CLASSIFICATION

## ABSTRACT

The spreading dynamics in social networks are often studied under the assumption that individuals' statuses, whether informed or infected, are fully observable. However, in many real-world situations, such statuses remain unobservable, which is crucial for determining an individual's potential to further spread the infection. While final statuses are hidden, intermediate indicators such as symptoms of infection are observable and provide useful representations of the underlying diffusion process. We propose a partial observability-aware Machine Learning framework to learn the characteristics of the spreading model. We term the method Distribution Classification, which utilizes the power of classifiers to infer the underlying transmission dynamics. Through extensive benchmarking against Approximate Bayesian Computation and GNN-based baselines, our framework consistently outperforms these state-of-the-art methods, delivering accurate parameter estimates across diverse diffusion settings while scaling efficiently to large networks. We validate the method on synthetic networks and extend the study to a real-world insider trading network, demonstrating its effectiveness in analyzing spreading phenomena where direct observation of individual statuses is not possible.

## 1 INTRODUCTION

Understanding the dynamics of spreading in networks is often challenging due to the absence of a comprehensive view of the connections between individuals involved in transmission. However, in today's increasingly digital environment, where user interactions, transactions, and communications are routinely logged and stored across platforms, reconstructing the transmission network is becoming more feasible (Zhou et al., 2017). The data may be noisy or incomplete, but the availability of large-scale digital traces offers a valuable foundation for inferring transmission pathways. Learning the transmission dynamics of contagion, whether in the context of disease, insider trading, or information spread, can be achieved by the network approach. It requires that the network structure is a meaningful substrate for the underlying transmission pathways (Dutta et al., 2018). When this condition is satisfied, diverse spreading phenomena can in theory be analyzed within a common analytical framework, enabling consistent modeling across different domains.

Traditional studies often assume full observability of individuals' transmission statuses, but in practice, such visibility is rarely available (Zhou et al., 2017; Pouget-Abadie & Horel, 2015; Newman, 2023; Wilinski & Lokhov, 2021). For example, during the COVID-19 pandemic, infection chains were frequently untraceable due to asymptomatic cases, misleading symptoms, and unreliable rapid tests. Similarly, in financial markets, the spread of private information through social connections is typically unobservable, making it difficult to identify who holds privileged information. In such scenarios, it remains unclear whether individuals were the carriers (either through being informed or infected) or when the transmission occurred. The absence of temporal and status information makes conventional Maximum Likelihood Estimation (MLE) methods unsuitable Gomez-Rodriguez et al. (2012). While recent research has started to address partially observed data Ramezani et al. (2023), existing methods fail to account for both hidden infection states and indirect symptom-based observations; in this work, we address this challenge, referred to as the Hidden Cascade (HC) problem.

Our method addresses the question of hidden or unreliable node status using symptom-based indirect observations in the context of cascades, and as such is a generalization of missing node status, as the nodes may exhibit false positive or false negative symptoms, not just missing observations. Hidden cascades are also related to cascade reconstruction from partial observations, a problem that has been widely studied. Existing approaches typically assume a one-sided observation model, where infected

nodes may be partially observed but uninfected nodes are never observed as infected. In contrast, we propose a two-sided observation model that also accounts for false positives, which makes the reconstruction task more challenging and highlights the novelty of our approach.

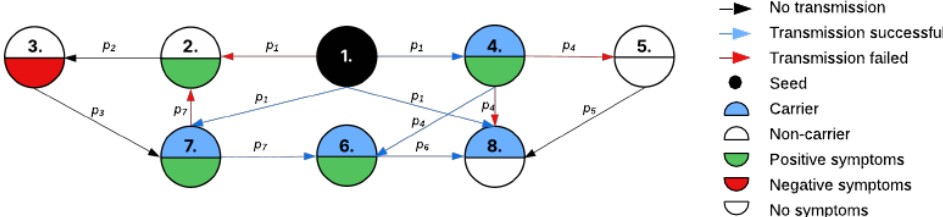

Figure 1: Illustration of Hidden Cascades (HCs).

Figure 1 illustrates the HCs, where the source of propagation is shown as a black node. In each cascade event, links between nodes are randomly activated with a propagation probabilities $p_i$, which represents the likelihood of propagation. We first assume a uniform propagation probability $p$ across all nodes, and this assumption is subsequently relaxed in Appendix C.1. We classify transmission links between nodes into three types: (1) successful transmissions (blue), where the target node is successfully infected by the source; (2) failed transmissions (red), where the source attempts transmission but the target does not receive it; and (3) non-transmissions (black), where the source node is not a carrier. The upper semicircle of each node indicates its status: blue represents an infected (carrier) node, while white denotes an non-carrier node. In HCs, the true status of individuals, such as whether they are carriers or not, is unobservable. Instead, observations are limited to symptoms, which are driven by the symptom probability $q$, applicable across both information and infection cascades. For tractability, our framework considers a uniform $q$ across all nodes, since allowing it to vary would substantially increase the complexity of estimation without contributing to the central focus of our study Gutmann et al. (2018). Positive symptoms, shown as green lower semicircles, are indicators that a node might be a carrier, but this cannot be confirmed with certainty. In information cascades, symptoms reflect behaviors influenced by the possible reception of information. For example, an investor connected to company insiders who makes a profitable trade before a public announcement, may exhibit behavior consistent with prior access to private information, although this is not definitive proof. A node can also be a carrier without displaying symptoms, referred to as an asymptomatic case (for example, node 8). On the other hand, symptoms may also be false positives, where a node shows signs that appear to indicate carrier status but are unrelated (as in node 2). Anti-symptoms, or negative symptoms, represented by red lower semicircles, may also occur. These suggest that a node is unlikely to be a carrier, though again not with certainty. Examples include an agent making a loss-making trade before a public announcement, suggesting they were likely not privately informed, or a person showing test results that contradict the usual disease profile. There is also a non-symptomatic state, where the individual shows no symptoms related to being a carrier. The overall transmission in the HCs is governed by two probabilities: the propagation probability $p$ and the symptom probability $q$.

The framework proposed in this paper is motivated by the principle of *distribution matching*, a technique that has been successfully applied across a variety of domains. Distribution matching enables synthetic data generation by preserving the statistical or receptive field properties of the original data, supporting tasks such as dataset condensation Hinton et al. (2015); Zhao & Bilen (2023) and graph condensation Liu et al. (2022). While distribution matching has demonstrated its versatility across various applications, this research introduces a novel framework termed *Distribution Classification* (DC). The proposed approach infers the underlying parameters of a *spreading model* $\psi(\theta)$ by classifying summary statistics, which serve as a holistic representation of the distributions from which the features are drawn. Rather than directly estimating the parameters, DC employs an adversarial strategy where a classifier is trained to be maximally uncertain in distinguishing between summary statistics generated from the ground truth parameter setting and those from sampled configurations, using this induced confusion as a mechanism for parameter inference.

The main contributions of this paper are multifold. 1) Problem Formulation of Hidden Cascades (HC): We introduce the Hidden Cascade problem, where individual infection statuses are unobservable and only indirect, noisy symptoms are available. This formulation generalizes classical cascade models

by incorporating partial and uncertain observations—an unexplored setting in network and machine learning literature. 2) Classifier-Based Inference via Distribution Classification: We propose a novel framework called Distribution Classification, which casts the inference of transmission parameters as a classification task. By training multiple entity-specific classifiers to distinguish between real and simulated data distributions, the method enables likelihood-free parameter recovery in complex, partially observed networks. 3) Comprehensive Experimental Validation: We evaluate the proposed approach on both synthetic networks (tree and loopy topologies) and a real-world insider trading network. The results demonstrate the robustness of the method in recovering transmission parameters under varying connectivity and noise conditions.

The paper is organized as follows: Section 2 introduces the concept of spreading processes in both epidemiological and financial contexts, along with relevant terminology and the overall framework. Section 3.1 describes the proposed methodology in detail. Section 3.2 describes the experimental setup used to evaluate the approach. In Section 4, we validate the proposed method using Monte Carlo simulations on synthetic data and discuss the results. Section 5 focuses on a real-world insiders network, presenting the learned company-specific parameters and corresponding analysis. Finally, Section 6 concludes the paper by summarizing the key findings. All experiments were conducted on a high-performance computing cluster using CPU nodes equipped with two Intel Xeon Gold 6230 "Cascade Lake" processors ($2 \times 20$ cores at 2.1 GHz) and 192 GiB of RAM per node.

## 2  RELATED WORK

**Epidemic Spread.** The study of epidemic spread has long been a prominent area of scientific inquiry and remains highly active to this day. One of the earliest and most notable contributions dates back to the 19[th] century, when John Snow investigated the spread of cholera in London, laying the foundation for modern epidemiology Snow (2023). Since then, extensive research has been dedicated to understanding how infectious diseases propagate through populations. In recent years, the spread of COVID-19 has been modeled probabilistically to capture its transmission dynamics Kucharski et al. (2020); Bherwani et al. (2021); Saxena et al. (2021). These models have supported vaccine and drug development and guided public health interventions. Classical models such as the *susceptible-infected* (SI) and *susceptible-infected-recovered* (SIR) frameworks Dutta et al. (2018) have been widely used to simulate and analyze such dynamics.

**Information Spread.** Beyond biological contagions, similar spreading processes appear in other domains, including social and financial networks. For instance, in financial settings, investors often share valuable non-public information with close contacts to enable profitable trades. Recent studies have used topological clustering and graph neural networks to identify individuals likely to receive such illicit information Goel et al. (2024); Baltakys et al. (2023), confirming the presence of hidden influence paths within these networks. These parallels between epidemiological and informational spreading highlight the broader utility of contagion modeling across domains.

**Existing methods.** A key challenge across both domains is reconstructing the underlying transmission pathways from limited observations. Maximum Likelihood Estimation (MLE) methods rely on fully observable timestamped data Gomez-Rodriguez et al. (2012), being unsuitable when only the final carrier statuses are known. To address it, prior work has explored inference strategies based on Message Passing methods, particularly the dynamic message passing (DMP) framework Lokhov (2016); Lokhov & Saad (2017); Wilinski & Lokhov (2021; 2024). While DMP performs well in sparse, tree-like networks, its accuracy diminishes in dense, loopy graphs, such as those found in financial networks, where short cycles violate its assumptions. Moreover, in many real-world settings, we observe only the final symptoms at the end of a time window, without access to the full temporal trajectory or the hidden infection state, defining the *Hidden Cascade (HC)* problem. These limitations have motivated us to develop the Hidden Cascade inference framework, which seeks to infer transmission dynamics from partial or noisy final observations. In such settings, observations may be indirect and uncertain, yet many existing approaches still assume full observability and certainty in carrier statuses. This mismatch restricts their applicability in real-world contexts, where observations are inherently partial and noisy.

In developing a framework for learning spreading processes through likelihood-free inference, we were inspired by the work of Gutmann et al. (2018), which introduced a likelihood-free inference framework based on classification. Methodologically, their framework differs from ours, particularly

because spreading processes on graphs require the use of distribution matching. While spreading processes have previously been inferred in a likelihood-free manner, the approaches used differ from the principles we adopt in this paper. Dutta et al. (2018) employed an approximate Bayesian computation method to simultaneously learn the parameters of the spreading process and identify the initially infected node. More recently, Wang & Onnela (2024) developed a Bayesian inference approach for estimating the parameters of a partially observed contagious process. Deep learning methods have also been explored; for example, Murphy et al. (2021) used deep Graph Neural Networks (GNN) to forecast the evolution of contagion dynamics.

## 3 PROPOSED METHOD

### 3.1 DISTRIBUTION CLASSIFICATION

We introduce a classification-based framework to infer the parameters governing *spreading processes* in complex systems. Rather than relying on global network statistics, we define an *entity-level distribution*, which captures individual behavior over time. These behaviors are shaped by a general *spreading model* $\psi(\theta)$, where contagion propagates across a system based on a set of governing parameters $\theta$. This approach aims to match the distribution of features extracted from real spreading data with those generated under simulated parameter settings. The core idea is to train a classifier that distinguishes between real and simulated data at the entity level based on their feature representations. The simulation parameters are iteratively adjusted to minimize the classifier's ability to distinguish between the two sources of data. Thus, the inference problem becomes one of Distribution Classification problem, with classification accuracy serving as a statistical discrepancy measure.

Let $\mathcal{D}_{\text{real}}^i = \{(x_j^i, y_i)\}_{j=1}^n$ and $\mathcal{D}_{\text{sim}}^{i,(\theta)} = \{(\tilde{x}_j^i, y_i)\}_{j=1}^n$ denote the datasets of real and simulated feature vectors for entity $i$, respectively, where $\theta \in \Theta$ represents the parameters of the spreading model $\psi$, $n$ is the number of feature vectors, and $y_i \in (0, 1)$ is the label. The real and simulated feature vectors are labeled as 1 and 0, respectively. The *entity-specific classification accuracy* is then defined as the measure of how effectively a classification model distinguishes between real and simulated feature vectors for entity $i$:

$$\text{CA}_i(\theta) = \mathbb{E}_{(x_i, y_i) \sim \mathcal{D}^{i,(\theta)}}[\mathbf{1}(f_\Phi^i(x_i) = y_i)],$$

where $f_\Phi^i$ is a classifier trained specifically for entity $i$. The *global classification accuracy*, which determines whether to accept the proposed parameters $\theta$ by measuring the overall discrepancy between real and simulated distributions, is obtained by averaging across all entities. The optimal parameters $\theta^*$ are estimated by minimizing the average classification accuracy, ensuring that real and simulated data are indistinguishable. By defining our framework in terms of a *general spreading process* $\psi(\theta)$ and a *flexible classifier* $f_\Phi$, our method accommodates various models of *disease transmission, financial contagion, and social influence*, making it broadly applicable to multiple domains. The inference process relies on three key components: the feature vectors associated with each entity, the optimized hyperparameters $\Phi$ for their respective classifiers, and the optimizer $\mathcal{O}$, which efficiently updates the model parameters $\theta$ during training.

### 3.2 EXPERIMENTAL SETUP

#### 3.2.1 MODELING HIDDEN CASCADES WITH THE INDEPENDENT CASCADE MODEL

In our setting, the underlying process generates the spread of infections or information over a network, resulting in cascades, where nodes become infected or informed. A *cascade* refers to the sequence of events across the network, capturing which entities become infectious and when. In this paper, however, the carrier statuses are latent and not directly observed. Instead, we observe indirect and noisy symptom signals emitted by the nodes, forming *Hidden Cascades*. These symptoms serve as indirect evidence: infected entities are likely to exhibit positive symptoms, while non-infected entities may still show symptoms spuriously, introducing ambiguity into the observed data. In this paper, the term infected is used broadly to refer both to epidemiological contagion and to the spread of information.

To model the true (but unobserved) spreading dynamics underlying these hidden cascades, we adopt the Independent Cascade (IC) model[1]. Let $G = (V, E)$ represent the network, where $V$ is the set of entities (nodes) and $E$ is the set of connections (edges). Each entity can either be a *carrier* (infected) or a *non-carrier* (not infected). At each time step $t$, an infected entity $i$ has a single chance to infect a neighboring entity $j$ at time $t + 1$, with a global propagation probability $p$. If this attempt fails, $j$ cannot be infected by $i$ again. Each entity may be infected at most once, and the cascade proceeds until no further infections are possible. Cascades are initiated from fixed seed sets (see Appendix A).

To simulate hidden cascades from the underlying infection processes, we introduce a probabilistic observation model based on symptom generation. Instead of directly observing the binary infection states of nodes, we observe noisy symptoms associated with each node. Each node $v \in V$ exhibits a symptom variable $z_v \in \{-1, 0, +1\}$, where $+1$ denotes a positive symptom, $-1$ a negative symptom, and $0$ indicates the absence of symptoms. The observed symptom $z_v$ is generated according to a conditional distribution based on the true (latent) infection state $a_v \in \{0, 1\}$, where $a_v = 1$ indicates infection and $a_v = 0$ otherwise.

Formally, the symptom distribution is defined as:

- For **infected nodes**[2] ($y_v = 1$):
  $$P(z_v = +1 \mid a_v = 1) = q, \quad P(z_v = 0 \mid a_v = 1) = 1 - q.$$
- For **non-infected nodes**[3] ($y_v = 0$):
  $$P(z_v = +1 \mid a_v = 0) = b_1, \quad P(z_v = -1 \mid a_v = 0) = b_2, \quad P(z_v = 0 \mid a_v = 0) = 1 - (b_1 + b_2) =: b_0.$$

As a result, the simulated data comprises symptom vectors instead of infection labels, introducing noise and ambiguity akin to real-world observations (see Appendix A for details).

### 3.2.2 FEATURE GENERATION FROM NOISY SYMPTOM OBSERVATIONS

Our goal is to construct robust node-level features from noisy, symptom-based observations of hidden cascades. For a given parameter set $\theta$, we simulate $N$ independent cascades. In each cascade $j \in \{1, \dots, N\}$, node $i$ emits a symptom observation $z_i^{(j)} \in \{-1, 0, +1\}$.

For each node, we compute the empirical distribution of its symptom values across $N$ cascades. Specifically, we define:

$$f_k = \frac{1}{N} \sum_{j=1}^{N} \mathbb{I}(z_i^{(j)} = k), \quad \text{for } k \in \{-1, 0, +1\}.$$

This yields three features per node: the fractions of positive, negative, and absent symptoms.

Unlike standard cascade models, hidden cascades lack explicit infection labels. Relying on raw cascade-level symptoms as direct learning targets is unreliable due to two major sources of noise:

**Stochastic Cascade Dynamics.** Infection events are governed by the probabilistic IC process, making each cascade realization inherently stochastic. A single cascade may not reflect the true influenceability of a node, especially for peripheral nodes rarely reached by the information.

**Exogenous Stochastic Events.** Nodes may emit false positives or anti-symptoms due to unrelated external processes. These exogenous signals introduce further noise that is not explained by the underlying spreading model.

To mitigate these issues, we aggregate symptoms across multiple cascade realizations, suppressing the impact of outlier behaviors and isolating persistent signal patterns.

To further model uncertainty in the diffusion process, we employ a *Monte Carlo*-based approach. For each node, we simulate $N$ independent cascades and compute symptom-based summary statistics. This procedure is repeated $M$ times, producing $M$ feature vectors per node. These samples

---

[1]The proposed method is agnostic to the specific model used to generate cascades. For our experiments, we adopt the Independent Cascade model.

[2]The probability of negative symptoms conditioned on infection is zero.

[3]We refer to this as Baseline Model.

collectively capture the distributional structure induced by the propagation and observation models, offering a compact yet expressive encoding of the node's diffusion behavior. Further implementation details, including pseudocode, are provided in Appendix D (Algorithm 1).

### 3.2.3 MODEL PARAMETER OPTIMIZATION

Due to the nature of our objective function, which is evaluated through Monte Carlo simulations, an analytical gradient is not available. The function is non-differentiable, making gradient-based optimization methods unsuitable. Consequently, we employ Powell's conjugate direction method, a derivative-free optimization algorithm that sequentially refines a set of search directions through one-dimensional minimizations Powell (1964). Rather than seeking a strict minimum directly, this method monitors changes in both the objective function (CA) and the parameter updates, using two tolerances: *function tolerance* ftol and *parameter tolerance* xtol. Convergence is achieved when the improvement in the objective function between successive iterations falls below ftol, and the maximum change in any parameter is less than xtol. Powell's method iteratively updates the search directions in the parameter space until these stopping criteria are met, ensuring stable and robust convergence. The learning procedure for optimal parameters with the associated algorithms is detailed in Appendix D (see Algorithm 2).

### 3.2.4 HYPERPARAMETER SELECTION FOR CLASSIFIER

While the goal is to train the model to exhibit maximum uncertainty in classification, it is equally important to ensure optimal performance. Hyperparameter optimization is essential to achieve the best settings for each classifier, balancing model accuracy with robustness under uncertainty. This ensures that the model not only captures data variability effectively but also performs efficiently in learning the propagation probabilities. In our approach, we train a separate classifier for each entity. Since the feature vectors depend on the transmission parameters $\theta$, every time a new $\theta$ is drawn from the parameter space $\Theta$, the feature vectors change—requiring the classifier to retune its hyperparameters. However, performing hyperparameter tuning at every iteration would be computationally expensive. To address this, we adopt the following strategy: hyperparameter tuning is performed only at the iteration where Powell's optimization converges. At this stage, for each entity, the classifier is fine-tuned using the feature vectors derived from the converged $\theta$. If the resulting classifier achieves higher classification accuracy than before, Powell's optimization is restarted using the previously converged $\theta$ as the new initial guess. The classifier tuning algorithm is provided in Appendix D (see Algorithm 3). The hyperparameter search space used for all selected classifiers is provided in Appendix E.

## 4 MONTE CARLO EXPERIMENTS

### 4.1 EVALUATION ON SYNTHETIC GRAPHS

We evaluated our method on two network structures: a *balanced tree* graph and a Barabási–Albert graph with $k = 2^4$ (see Figure 3a and 3b in Appendix F ). Initially, only one entity (the seed) is the infected at time $t = 0$, acting as the carrier that begins transmitting to its neighboring entities over subsequent time points. We tested multiple classifiers, including Decision Tree, K-Nearest Neighbors (KNN), Logistic Regression, Naive Bayes, Random Forest, Stochastic Gradient Descent (SGD), and Support Vector Machine (SVM). In addition to the summary statistics discussed in Section 3.2.2, which are considered an optimal feature set, we extend this set by including the mean, variance, entropy, and the changes in the number of positive, negative, and no-symptom cases between earlier and later time intervals, thereby capturing temporal variation in symptom occurrence. This extension aimed to capture temporal variations in symptom progression. From Table 1, it can be confirmed that the SVM classifier, with its optimal hyperparameter setting, outperforms the other classifiers, and that the optimal feature set more effectively captures the underlying spreading model's parameters; hence, we fix both the classifier and the summary statistic for all subsequent experiments. To compare the distribution of actual and simulated data under the learned parameters, we split nodes by distance from the seed and by degree. Figure 2 shows that the symptom distribution in the simulated cascades

---

[4]$k = 2$ leads to a graph with a high density of connections and the presence of loops, as each new node connects to only a few existing nodes, rapidly creating densely interconnected structures.

closely matches that of the real data. Nodes closer to the seed and those with higher degree tend to exhibit more positive symptoms, capturing key structural effects observed in real cascades.

Table 1: Comparison of classifier performance across summary statistics in *tree* and *loopy* graph structures. Both setups use propagation probability $p = 0.3$, symptom probability $q = 0.7$.

| Graph Type | Classifier | Reduced Summary Statistic | | | | Extended Summary Statistic | | | |
|---|---|---|---|---|---|---|---|---|---|
| | | $\hat{p}$ | $\hat{q}$ | $\overline{MSE}$ | $\overline{CA}$ | $\hat{p}$ | $\hat{q}$ | $\overline{MSE}$ | $\overline{CA}$ |
| | Decision Tree | $0.26 \pm 0.07$ | $0.74 \pm 0.08$ | $6.16 \times 10^{-3}$ | 0.49 | $0.26 \pm 0.03$ | $0.71 \pm 0.09$ | $5.24 \times 10^{-3}$ | 0.48 |
| | KNN | $0.28 \pm 0.05$ | $0.77 \pm 0.10$ | $7.61 \times 10^{-3}$ | 0.49 | $0.22 \pm 0.14$ | $0.39 \pm 0.35$ | $1.09 \times 10^{-1}$ | 0.48 |
| | Logistic Regression | $0.95 \pm 0.03$ | $0.94 \pm 0.11$ | $2.43 \times 10^{-1}$ | 0.43 | $0.21 \pm 0.09$ | $0.68 \pm 0.04$ | $8.37 \times 10^{-3}$ | 0.46 |
| Tree | Naive Bayes | $0.30 \pm 0.03$ | $0.72 \pm 0.04$ | $1.10 \times 10^{-3}$ | 0.47 | $0.25 \pm 0.09$ | $0.58 \pm 0.27$ | $4.17 \times 10^{-2}$ | 0.49 |
| | Random Forest | $0.24 \pm 0.08$ | $0.73 \pm 0.03$ | $4.99 \times 10^{-3}$ | 0.48 | $0.30 \pm 0.02$ | $0.71 \pm 0.06$ | $1.74 \times 10^{-3}$ | 0.48 |
| | SGD | $0.27 \pm 0.04$ | $0.70 \pm 0.04$ | $1.67 \times 10^{-3}$ | 0.47 | $0.29 \pm 0.18$ | $0.64 \pm 0.09$ | $1.79 \times 10^{-2}$ | 0.47 |
| | SVM | $0.31 \pm 0.01$ | $0.67 \pm 0.03$ | $1.04 \times 10^{-3}$ | 0.43 | $0.28 \pm 0.03$ | $0.73 \pm 0.02$ | $1.18 \times 10^{-3}$ | 0.43 |
| | Decision Tree | $0.30 \pm 0.02$ | $0.73 \pm 0.09$ | $3.86 \times 10^{-3}$ | 0.51 | $0.26 \pm 0.09$ | $0.68 \pm 0.02$ | $4.21 \times 10^{-3}$ | 0.58 |
| | KNN | $0.22 \pm 0.11$ | $0.46 \pm 0.33$ | $8.00 \times 10^{-2}$ | 0.67 | $0.28 \pm 0.04$ | $0.69 \pm 0.06$ | $2.25 \times 10^{-3}$ | 0.48 |
| | Logistic Regression | $0.47 \pm 0.15$ | $0.38 \pm 0.00$ | $7.46 \times 10^{-2}$ | 0.43 | $0.28 \pm 0.01$ | $0.73 \pm 0.10$ | $4.46 \times 10^{-3}$ | 0.46 |
| Loopy | Naive Bayes | $0.22 \pm 0.11$ | $0.47 \pm 0.33$ | $8.01 \times 10^{-2}$ | 0.67 | $0.18 \pm 0.11$ | $0.34 \pm 0.33$ | $1.20 \times 10^{-1}$ | 0.78 |
| | Random Forest | $0.25 \pm 0.09$ | $0.63 \pm 0.31$ | $4.45 \times 10^{-2}$ | 0.61 | $0.24 \pm 0.13$ | $0.70 \pm 0.02$ | $7.67 \times 10^{-3}$ | 0.57 |
| | SGD | $0.17 \pm 0.10$ | $0.52 \pm 0.39$ | $8.93 \times 10^{-2}$ | 0.75 | $0.27 \pm 0.03$ | $0.75 \pm 0.10$ | $6.03 \times 10^{-3}$ | 0.48 |
| | SVM | $0.30 \pm 0.00$ | $0.69 \pm 0.0047$ | $4.36 \times 10^{-5}$ | 0.42 | $0.30 \pm 0.07$ | $0.70 \pm 0.02$ | $1.93 \times 10^{-4}$ | 0.43 |

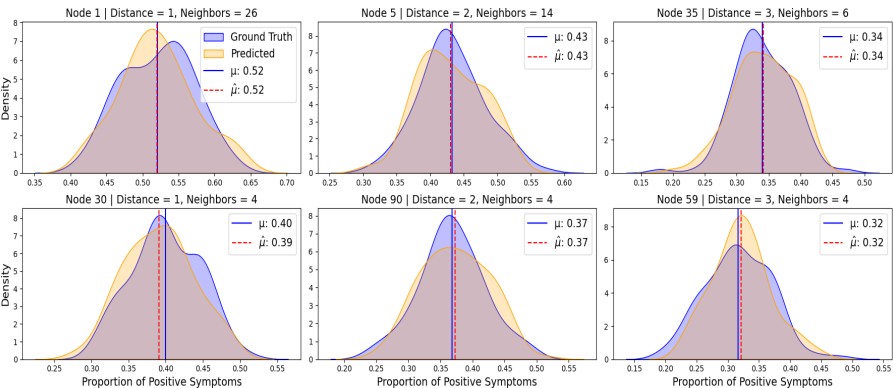

Figure 2: Distribution of positive symptoms in a loopy graph, organized by both distance from the seed node and node connectivity. Two key observations can be made. **Influence of Distance from the Seed Node:** Nodes closer to the seed entity (i.e., within the 1-hop neighborhood) have a higher likelihood of being infected. This is evident when moving from left to right in the figure, where nodes are arranged in increasing order of distance. The distribution becomes increasingly skewed as the distance increases, indicating a decline in the proportion of positive symptoms. **Effect of Connectivity:** Nodes with higher connectivity (i.e., more neighbors) have a higher chance of infection. This is observable when comparing nodes from top to bottom in the figure, where the first-row nodes have more neighbors than those in the second row. The simulated distributions, generated using the inferred parameters, closely match the actual data, confirming that the model captures key structural effects in the spreading process.

## 4.2 EVALUATION ON EMPIRICAL SOCIAL GRAPH (INSIDERS NETWORK)

The proposed method is subsequently validated using the insiders' network, which is described in detail in Section 5.1. In this setup, rather than generating ground truth parameters $\theta$ from transaction data, we fix the ground truth parameters *a priori*. The objective is to evaluate the model's effectiveness in a larger, more complex network with a greater number of entities. Unlike previous setups, this experiment involves multiple seed entities. The spreading process begins from each seed entity sequentially, with an incremental time offset. Specifically, the first seed entity initiates spreading at time $t$, the second at time $t + 1$, and so on. As shown in Figure 3c in Appendix F, the nodes marked in red represent these seed entities, which are distributed across different regions of the network to ensure extensive spread.

### 4.3 BENCHMARKING, ROBUSTNESS AND EFFICIENCY ANALYSIS

First, we benchmarked our method against four alternative approaches in Appendix B: i) the approach proposed by Gutmann et al. (2018) and three Graph Neural Network Architectures. Our distribution-matching approach consistently achieved lower errors and more stable parameter recovery. To evaluate the framework's performance and efficiency, we conducted several robustness checks in Appendix C. First, we relax the assumption of a uniform propagation probability $p$ and examine robustness under node-specific propagation probabilities. The results, reported in Appendix C.1, show that our method remains effective. Second, in Appendix C.2, we evaluate performance under both the Independent Cascade and Threshold models, and again our approach proves effective. Third, in Appendix C.3 we tested robustness across diverse network topologies, including tree, loopy, and empirical insider trading networks, while varying the spreading model parameters $p$ and $q$ over the set $\{0.1, 0.3, 0.5, 0.7, 0.9\}$, learning the corresponding estimates $\hat{p}$ and $\hat{q}$ for each combination, and observed consistent accuracy and parameter recovery across all settings. Finally, to analyze computational efficiency, in Appendix C.4 we evaluated scalability on synthetic BA graphs, and find that runtime grows sublinearly to slightly superlinearly with network size. These results demonstrate that our framework generalizes well, scales efficiently, and remains robust across different settings, diffusion models and network topologies.

## 5 EMPIRICAL ANALYSIS

To validate the proposed method beyond synthetic data and known ground truth i.e. model's parameters, we conduct an extensive empirical analysis using real-world empirical data on (partially) observable social links between directors, classified as insiders and their transactions across all securities, even those outside insider trading regulations. We hypothesize that information can begin spreading privately through a company's board's social links before an official announcement, with further propagation halting once the information becomes public. The dataset comprises transaction records from 28 companies, including details of their respective board members and investors and all of their transactions [5]. For each company announcement, we define the *Pre-Announcement Period* as the four days preceding the disclosure. For simplicity, we refer to this as the *announcement period*. Our primary objective is to evaluate whether *information transmission*, as inferred by the method, is significantly higher during the *announcement periods* compared to the *Non-Announcement periods*, which include all other trading days. This would align with our hypothesis regarding the private dissemination of forthcoming information prior to its public disclosure. This section describes the empirical dataset, the process of extracting *feature vectors* for ground truth comparison, and discusses the results of the findings.

### 5.1 INSIDERS' DATA

Individuals who trade in a specific stock are referred to as *agents*, and the network formed through their social connections is the *insiders network*. This network includes agents who serve as board members of one or more companies. These individuals are believed to possess valuable insider information regarding the future direction of stock prices Goel et al. (2024). Those who trade based on such information reportedly achieve substantial returns—approximately 35% over a 21-day period Ahern (2017). In our empirical analysis, we assume that information disseminates within this network prior to the release of public company announcements, triggering a cascade of opportunistic trading. When insider information becomes available, agents are assumed to act on it by executing profitable buy or sell trades. The hypothesis is that information may begin to spread from a company's board through social links prior to a public announcement. To analyze this hypothesis, we utilize a unique and comprehensive dataset compiled from multiple sources. i) Insider network data in Finland: Being an insider in the same company establishes a social link among all members. Overlapping board memberships consequently form a large, interconnected social network. ii) Board member information and insider trading disclosures: Mandatory disclosure notifications by insiders enable us to identify partial trading patterns. iii) Pseudonymized trading data: A unique and extensive dataset containing detailed trading records of all investors across all securities listed on the Helsinki Stock Exchange. iv) Company announcement data: Information regarding the timing and nature of public company disclosures (See Tables 11 and 12 in Appendix G for descriptive statistics).

---

[5]We have an agreement with Euroclear Finland that grants us access rights to the data.

## 5.2 RESULTS

In the context of the *insider trading* network, each spreading cascade corresponds to an *information cascade* that precedes a public announcement. For a company issuing $n$ public announcements, this results in $n$ *independent information cascades*. Within each cascade, the participating entities are investors, and their observable *symptoms* are defined based on trading outcomes: a *positive symptom* indicates a profitable trade, a *negative symptom* denotes a loss, and the absence of a symptom represents no trading activity. These outcomes collectively form a *trade matrix*, which captures investor behavior across cascades in a structured format. The goal is to infer the spreading model's parameters: $p$ and $q$. Prior to inference, it is necessary to extract the ground truth feature vectors from the trade data. Algorithm 4 in Appendix D details the procedure used to construct the feature set $\mathcal{F}_{\text{GT}}$. All subsequent experiments adhere to the setup described in Appendix A. For each company, the $p$ and $q$ probabilities are estimated separately for the announcement and non-announcement periods.

Table 14 (see Appendix G) presents the average inferred parameters ($\hat{p}$ and $\hat{q}$) across a set of companies, comparing the announcement and non-announcement periods. The non-announcement window is intended to serve as a baseline, under the assumption that information transmission within insider networks should be minimal in the absence of public news. If investors made their decisions completely independently, the parameters $p$ and $q$ during these periods should be close to zero. However, the data reveals a different story. Both $\hat{p}_n$ and $\hat{q}_n$ are frequently non-zero and, in several cases, considerably high. This is because the baseline model does not account for investors' collective responses to exogenous factors, such as macroeconomic news and market volatility. These factors can largely explain the observed co-occurrences in their trading activity and trigger synchronized trades across the stock market Baltakiene et al. (2021). As a consequence, the parameter estimates $\hat{p}$ are expected to be positive even outside of the announcement periods, serving as a useful comparative reference for evaluating parameter estimates between announcement and non-announcement periods.

A consistent pattern emerges: propagation probabilities tend to be even higher during announcement periods. For most companies, $\hat{p}_a > \hat{p}_n$, likely linked to the dissemination of private information. For example, Company 3 shows a $p$-ratio of 1.86, while Company 1 has a ratio of 2.09, highlighting the heightened flow of information in these windows. There are notable exceptions. Companies such as Company 26, Company 27, and Company 28 display lower $\hat{p}$ values in announcement periods than in non-announcement periods. These anomalies may reflect scenarios where insiders acted in advance of the announcement, or where market-moving information emerged from other sources. Alternatively, they may be due to company-specific factors such as internal trading restrictions or more cautious trading behavior during high-scrutiny periods. The results consistently indicates higher propagation probabilities in announcement periods, reinforcing the view that these windows are associated with greater information flow.

## 6 CONCLUSION

We introduce a model-agnostic, classification-driven framework for inferring parameters of spreading processes from noisy, symptom-based observations. By framing inference as a distribution classification problem, we estimate parameters such that simulated and observed data become indistinguishable to the classifier. The framework flexibly supports diverse spreading models and classifier architectures, enabling broad applicability. Empirically, the method consistently recovers diffusion parameters across varying values of $p$ and $q$ on synthetic as well as real graphs, demonstrating robustness to different underlying spreading dynamics. On the empirical insider trading network with transaction-level data, the method infers firm-level information spreading dynamics. In particular, the estimated spreading information probability values ($p$) are consistently higher during announcement periods for most firms, suggesting that insiders spread tips about forthcoming announcements prior to their official publication. Beyond methodological contributions, the framework yields domain-specific insights. In finance, it helps supervisory authorities prioritize market surveillance and target regulations toward specific companies and in epidemic modeling, for instance, estimating city-level $p$ and $q$ allows identification of transmission hubs and prioritization of containment efforts. The ability to operate without full infection labels, yet recover interpretable parameters, makes this approach well suited for real-world scenarios where observation is indirect and noisy.

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

## A  EXPERIMENT SETUP

1. **Graph**: Let $G = (V, E)$ be a directed graph, where $V$ represents the set of entities, and $E$ denotes the set of directed edges representing connections between them. The total number of entities, denoted by $N$, is equal to the number of nodes in the graph.

2. **Parameters to be inferred:** The spreading process within the network is governed by two critical parameters. The first, the propagation probability ($p$), is assigned to each directed edge, representing the likelihood that an entity will disseminate infection or information to others within the network. The second, the symptom probability ($q$), quantifies the chance that an entity will exhibit a positive symptom after receiving it.

3. **Baseline Model:** In the *Monte Carlo experiments*, the baseline symptom behavior for non-infected entities is defined by a fixed probability vector $b = [b_i]_{i=0}^2 = \{0.5, 0.25, 0.25\}$, representing the probabilities of showing no symptom, a positive symptom, and a negative symptom, respectively. In the *empirical experiment*, we define baseline symptom distributions for each investor based on their trading activity during periods with no company-specific information. Let $n_x$ denote the total number of trading days in the non-announcement window for company $x$. For each investor $i$, we compute the baseline symptom vector:

$$b_i = \{b_0^i,\ b_1^i,\ b_2^i\} = \left\{ \frac{n_i^0}{n_x},\ \frac{n_i^+}{n_x},\ \frac{n_i^-}{n_x} \right\},$$

where $n_i^0$, $n_i^+$, and $n_i^-$ correspond to the number of no-trade days, profitable trade days, and loss-making trade days, respectively, for investor $i$ during the non-announcement period.

4. **Independent Cascade (IC) Model:** Entities that have received information or have been infected are considered to be in an *infected* state. To generate an independent cascade (IC) model with a single seed entity, we follow the procedure below to obtain a list of infected entities:

   (a) Initialize the seed entities.
   (b) Define the state of each entity as not infected.
   (c) Set the state for the seed entities (the initial set of infected entities) as infected.
   (d) Mark all the edges (transmission links) as "not yet tried." This means that these edges have not been used for propagation.
   (e) To generate the independent cascade model, we define a set of infected entities, which are the newly infected entities in each iteration (for the first iteration, the seed entities are infected). While the set of infected entities is non-empty, we perform the following steps:
      i. For every entity in the set of infected entities, we observe all the 1-hop neighbors with an outward edge. These are the entities that are directly connected to the infected entities.
      ii. If the edge state is "not yet tried," a random number $r$ is generated in the range $[0, 1]$. The neighboring entity is set as infected if the edge weight $P$ is greater than $r$, and the edge state is then updated to "tried." This indicates that the edge has been used for propagation and cannot be utilized again.
      iii. Finally, we update the set of infected entities by replacing the initial set with the set of entities infected in this iteration. We continue this process until the set of infected entities is empty.
   In the end, we obtain the state of each entity as either infected or not infected.

5. **Assignment of symptoms:** When an entity is infected, we assume that it will exhibit a positive symptom. However, randomness is introduced into this process—when considering an insider network, an informed investor will either decide to use the information to trade profitably or choose not to trade at all, but never engage in a non-profitable trade. On the other hand, if an entity is not infected, symptoms are determined based on a predefined baseline model.

## B  BENCHMARKING AGAINST ALTERNATIVE MODELS

We benchmark our framework against the method proposed in Gutmann et al. (2018), which uses Approximate Bayesian Computation (ABC) for likelihood-free inference. A key feature of this approach is that the discrepancy function is defined independently of the internal cascade dynamics, making it applicable in settings with limited temporal information. To ensure a fair comparison, we implemented Gutmann et al. (2018) under the same experimental conditions as our framework. The results in Table 2 demonstrate that Gutmann et al. (2018) performs poorly in this setting, with the estimated propagation probabilities ($\hat{p}$) and symptom probabilities ($\hat{q}$) exhibiting substantial variance and notable deviations from the ground-truth values. The issue is particularly pronounced in Loopy graphs, where the Mean Squared Error (MSE) reaches $2.56 \times 10^{-1}$, underscoring the method's sensitivity to noisy observations. In contrast, our framework consistently yields parameter estimates close to the true values and achieves markedly lower MSE across both Tree and Loopy structures (see Table 7 and 8 ).

Table 2: Performance of the method in Gutmann et al. (2018) under the IC model with symptom probability $q = 0.7$. Results are shown for Tree and Loopy network topologies. Here, $p$ denotes the true propagation probability, and $\hat{p}$, $\hat{q}$ are the estimated parameters.

| | Tree | | | | Loopy | | | |
|---|---|---|---|---|---|---|---|---|
| $p$ | $\hat{p}$ | $\hat{q}$ | $\overline{MSE}$ | $\overline{CA}$ | $\hat{p}$ | $\hat{q}$ | $\overline{MSE}$ | $\overline{CA}$ |
| 0.1 | $0.33 \pm 0.141$ | $0.46 \pm 0.299$ | $9.11 \times 10^{-2}$ | 0.35 | $0.28 \pm 0.128$ | $0.58 \pm 0.214$ | $5.23 \times 10^{-2}$ | 0.37 |
| 0.3 | $0.25 \pm 0.027$ | $0.43 \pm 0.224$ | $5.55 \times 10^{-2}$ | 0.38 | $0.29 \pm 0.092$ | $0.53 \pm 0.221$ | $4.18 \times 10^{-2}$ | 0.38 |
| 0.5 | $0.73 \pm 0.129$ | $0.69 \pm 0.006$ | $3.27 \times 10^{-2}$ | 0.40 | $0.71 \pm 0.107$ | $0.69 \pm 0.022$ | $2.65 \times 10^{-2}$ | 0.42 |
| 0.7 | $0.63 \pm 0.097$ | $0.70 \pm 0.006$ | $5.38 \times 10^{-3}$ | 0.36 | $0.42 \pm 0.365$ | $0.51 \pm 0.345$ | $1.42 \times 10^{-1}$ | 0.59 |
| 0.9 | $0.62 \pm 0.230$ | $0.63 \pm 0.197$ | $8.57 \times 10^{-2}$ | 0.46 | $0.41 \pm 0.356$ | $0.45 \pm 0.318$ | $2.56 \times 10^{-1}$ | 0.64 |

We additionally benchmark our model with several well-known Graph Neural Network (GNN) variants, including Graph Attention Network(GAT), alongside Graph Convolutional Network(GCN)-based architectures, and report the best results. While GNN-based approaches offer considerable modeling flexibility, their effective deployment involves non-trivial design choices such as selecting the appropriate GNN variant, tuning embedding dimensions, and determining network depth. These steps are computationally demanding and require substantial amounts of data for training.

For comparison, we implemented a two-layer GAT, two-layer GCN and four-layer GCN and evaluated it under the same experimental setting. The model is trained over the grid $[0, 1]$ with a step size of 0.1 for both $p$ and $q$, and its architecture can be expressed as:

The GCN and GAT architectures used in our experiments are summarized in the Table 3. Each model maps node features $x$ to the estimated propagation and symptom probabilities $[\hat{p}, \hat{q}]$.

Table 3: Comparison of Graph Neural Network Architectures.

| Architecture | Layers | Activation | Pooling | Output |
|---|---|---|---|---|
| 2-layer GAT | GAT $\rightarrow$ GAT | ReLU | GlobalMeanPool | Sigmoid |
| 2-layer GCN | GCN $\rightarrow$ GCN | ReLU | GlobalMeanPool | Sigmoid |
| 4-layer GCN | GCN $\rightarrow$ GCN $\rightarrow$ GCN $\rightarrow$ GCN | ReLU | GlobalMeanPool | Sigmoid |

Across all variants, the GNN-based models underperformed relative to our distribution-matching (DC) approach, particularly for Loopy graphs, where high standard deviation and elevated mean squared error (MSE) were observed. In contrast, our framework delivers robust parameter estimates without requiring supervised training, making it significantly more practical for data-scarce scenarios. Detailed numerical results are reported in Table 4.

These observations justify our methodological choice: while GNNs present an interesting alternative, the additional complexity and training requirements did not yield improved performance in our experimental setting. Our distribution-matching framework aligns the feature distributions of observed and simulated data, eliminating the need for large labeled datasets while maintaining reliable performance. Exploring advanced GNN architectures and training strategies remains a promising avenue for future research.

Table 4: Comparison of parameter estimation across multiple models (GAT, GCN, and GCN 4-layer) for Tree and Loopy graphs under the IC model ($q = 0.7$ fixed). Each cell shows mean ± standard deviation over 5 samples, and MSE is reported in scientific notation.

| | Tree | | | Loopy | | |
|---|---|---|---|---|---|---|
| $p$ | $\hat{p}$ | $\hat{q}$ | $\overline{MSE}$ | $\hat{p}$ | $\hat{q}$ | $\overline{MSE}$ |
| | | | 2-layer GAT | | | |
| 0.1 | $0.20 \pm 0.020$ | $0.67 \pm 0.035$ | $6.69 \times 10^{-3}$ | $0.13 \pm 0.006$ | $0.44 \pm 0.087$ | $3.81 \times 10^{-2}$ |
| 0.3 | $0.28 \pm 0.042$ | $0.73 \pm 0.039$ | $2.40 \times 10^{-3}$ | $0.32 \pm 0.011$ | $0.62 \pm 0.025$ | $3.58 \times 10^{-3}$ |
| 0.5 | $0.47 \pm 0.042$ | $0.81 \pm 0.025$ | $8.22 \times 10^{-3}$ | $0.51 \pm 0.028$ | $0.65 \pm 0.004$ | $1.54 \times 10^{-3}$ |
| 0.7 | $0.72 \pm 0.012$ | $0.84 \pm 0.070$ | $1.29 \times 10^{-2}$ | $0.68 \pm 0.023$ | $0.67 \pm 0.004$ | $1.03 \times 10^{-3}$ |
| 0.9 | $0.91 \pm 0.008$ | $0.73 \pm 0.005$ | $4.16 \times 10^{-4}$ | $0.89 \pm 0.006$ | $0.65 \pm 0.010$ | $1.14 \times 10^{-3}$ |
| | | | 2-layer GCN | | | |
| 0.1 | $0.17 \pm 0.034$ | $0.59 \pm 0.094$ | $1.36 \times 10^{-2}$ | $0.08 \pm 0.006$ | $0.67 \pm 0.040$ | $1.50 \times 10^{-3}$ |
| 0.3 | $0.33 \pm 0.055$ | $0.70 \pm 0.057$ | $3.56 \times 10^{-3}$ | $0.29 \pm 0.007$ | $0.70 \pm 0.014$ | $1.56 \times 10^{-4}$ |
| 0.5 | $0.50 \pm 0.048$ | $0.73 \pm 0.047$ | $2.78 \times 10^{-3}$ | $0.49 \pm 0.007$ | $0.70 \pm 0.009$ | $1.09 \times 10^{-4}$ |
| 0.7 | $0.71 \pm 0.012$ | $0.72 \pm 0.034$ | $9.16 \times 10^{-4}$ | $0.66 \pm 0.011$ | $0.70 \pm 0.011$ | $1.04 \times 10^{-3}$ |
| 0.9 | $0.91 \pm 0.009$ | $0.68 \pm 0.013$ | $4.96 \times 10^{-4}$ | $0.90 \pm 0.005$ | $0.69 \pm 0.004$ | $9.50 \times 10^{-5}$ |
| | | | 4-layer GCN | | | |
| 0.1 | $0.17 \pm 0.045$ | $0.41 \pm 0.104$ | $4.98 \times 10^{-2}$ | $0.06 \pm 0.002$ | $0.75 \pm 0.063$ | $3.93 \times 10^{-3}$ |
| 0.3 | $0.23 \pm 0.016$ | $0.82 \pm 0.025$ | $9.81 \times 10^{-3}$ | $0.29 \pm 0.013$ | $0.66 \pm 0.011$ | $1.20 \times 10^{-3}$ |
| 0.5 | $0.45 \pm 0.046$ | $0.75 \pm 0.062$ | $5.62 \times 10^{-3}$ | $0.45 \pm 0.034$ | $0.69 \pm 0.011$ | $1.96 \times 10^{-3}$ |
| 0.7 | $0.67 \pm 0.007$ | $0.72 \pm 0.027$ | $9.96 \times 10^{-4}$ | $0.71 \pm 0.059$ | $0.69 \pm 0.006$ | $1.82 \times 10^{-3}$ |
| 0.9 | $0.89 \pm 0.012$ | $0.74 \pm 0.006$ | $1.07 \times 10^{-3}$ | $0.91 \pm 0.004$ | $0.69 \pm 0.008$ | $1.47 \times 10^{-4}$ |

## C  ROBUSTNESS AND EFFICIENCY ANALYSIS

### C.1  NODE-SPECIFIC PROPAGATION PROBABILITIES

We conducted robustness experiments by sampling $p \sim U(0, 1)$ with fixed $q$ and further compared our framework against a two-layer GCN model that outputs node-specific $p$. Table 5 summarizes the results across three propagation probability ranges: $(0, 1)$, $(0, 0.5)$, and $(0.5, 1)$, with actual $p$ uniformly distributed in each range. The GCN performs reasonably well for $p \in (0, 0.5)$ but its accuracy deteriorates sharply for higher propagation values, failing almost entirely when $p \in (0.5, 1)$. In contrast, DC maintains stable accuracy and low error across all ranges, demonstrating robustness under both low and high propagation probabilities. The slightly lower accuracy and higher MSE observed for DC in the $(0, 1)$ case are expected, as this setting spans the full propagation range and is inherently more challenging than the narrower intervals. The number of assignments was kept consistent across all settings. These findings reveal a key limitation of GCN-based models in handling heterogeneous propagation dynamics, whereas the proposed distribution-matching approach remains effective across regimes. Nevertheless, due to the stochastic nature of diffusion processes, perfectly estimating node-specific $p$ or $q$ values is not feasible.

Table 5: Performance of DC vs. GCN models ($q = 0.7$) under different propagation probability ranges. Metrics: MSE, Acc@0.1, Acc@0.2. Acc@0.1 measures the proportion of nodes whose predicted parameter lies within $\pm 0.1$ of the true value (similarly for Acc@0.2). Each GCN experiment uses 1000 assignments with an 80-10-10 split.

| Range of $p$ and $\hat{p}$ | Model | MSE | Acc@0.1 | Acc@0.2 |
|---|---|---|---|---|
| $(0, 1)$ | GCN | 0.1444 | 0.20 | 0.39 |
| | DC | 0.1401 | 0.25 | 0.47 |
| $(0, 0.5)$ | GCN | 0.0328 | 0.39 | 0.69 |
| | DC | 0.0362 | 0.41 | 0.69 |
| $(0.5, 1)$ | GCN | 0.5274 | 0.01 | 0.06 |
| | DC | 0.0362 | 0.41 | 0.69 |

## C.2    ROBUSTNESS TO DIFFUSION MODEL VARIATIONS

Although our main experiments focused on the IC model, the proposed learning framework is model-agnostic and can be applied with any diffusion process, provided the propagation dynamics are specified. To illustrate this, we additionally evaluated our method under the Linear Threshold model Granovetter (1978). The results under LT closely mirror those obtained for IC, yielding comparable parameter recovery and predictive performance (see Table 6). These findings confirm that our approach remains robust across different diffusion mechanisms, while exploration of further models is left for future work.

Table 6: Evaluation under the LT model with symptom probability $q = 0.7$. Here, $p_{thr}$ denotes the activation threshold (fraction of neighbors required for activation), and $\hat{p}_{thr}$ is its estimated value.

| | Tree | | | | Loopy | | | |
|---|---|---|---|---|---|---|---|---|
| $p_{thr}$ | $\hat{p}_{thr}$ | $\hat{q}$ | $\overline{MSE}$ | $\overline{CA}$ | $\hat{p}_{thr}$ | $\hat{q}$ | $\overline{MSE}$ | $\overline{CA}$ |
| 0.1 | $0.18 \pm 0.007$ | $0.78 \pm 0.015$ | $1.38 \times 10^{-2}$ | 0.43 | $0.19 \pm 0.064$ | $0.69 \pm 0.057$ | $3.66 \times 10^{-3}$ | 0.43 |
| 0.3 | $0.27 \pm 0.035$ | $0.61 \pm 0.010$ | $9.99 \times 10^{-3}$ | 0.43 | $0.30 \pm 0.127$ | $0.71 \pm 0.078$ | $1.25 \times 10^{-5}$ | 0.43 |
| 0.5 | $0.57 \pm 0.028$ | $0.70 \pm 0.002$ | $6.32 \times 10^{-3}$ | 0.42 | $0.60 \pm 0.064$ | $0.73 \pm 0.037$ | $4.85 \times 10^{-3}$ | 0.42 |
| 0.7 | $0.68 \pm 0.026$ | $0.70 \pm 0.004$ | $1.44 \times 10^{-3}$ | 0.43 | $0.79 \pm 0.057$ | $0.70 \pm 0.000$ | $4.05 \times 10^{-3}$ | 0.42 |
| 0.9 | $0.88 \pm 0.039$ | $0.70 \pm 0.002$ | $1.15 \times 10^{-3}$ | 0.43 | $0.86 \pm 0.007$ | $0.74 \pm 0.000$ | $1.81 \times 10^{-3}$ | 0.42 |

## C.3    ROBUSTNESS TO NETWORK TOPOLOGY

To further examine the stability of our approach, we conducted simulation experiments on synthetic and empirical networks with known ground truth (Tables 7, 8, and 9). These experiments systematically varied the propagation probability $p$ and the symptom probability $q$, and were carried out on three distinct graph structures: tree, loopy, and empirical topologies. Importantly, these evaluations focus exclusively on the structural properties of the networks, independent of investor behavior or transaction data.

Table 7: Robustness Check for Tree Across Multiple Values of $p$ and $q$

| | | q (0.1) | | | | q (0.3) | | | |
|---|---|---|---|---|---|---|---|---|---|
| | | $\hat{p}$ | $\hat{q}$ | $\overline{MSE}$ | $\overline{CA}$ | $\hat{p}$ | $\hat{q}$ | $\overline{MSE}$ | $\overline{CA}$ |
| | 0.1 | $0.13 \pm 0.040$ | $0.10 \pm 0.000$ | $1.20 \times 10^{-3}$ | 0.43 | $0.15 \pm 0.030$ | $0.36 \pm 0.140$ | $1.17 \times 10^{-2}$ | 0.43 |
| | 0.3 | $0.29 \pm 0.020$ | $0.12 \pm 0.020$ | $7.03 \times 10^{-4}$ | 0.43 | $0.28 \pm 0.030$ | $0.31 \pm 0.020$ | $7.62 \times 10^{-4}$ | 0.43 |
| $p$ | 0.5 | $0.46 \pm 0.060$ | $0.10 \pm 0.000$ | $2.22 \times 10^{-3}$ | 0.43 | $0.49 \pm 0.007$ | $0.29 \pm 0.020$ | $2.94 \times 10^{-4}$ | 0.43 |
| | 0.7 | $0.70 \pm 0.005$ | $0.10 \pm 0.000$ | $1.62 \times 10^{-5}$ | 0.43 | $0.70 \pm 0.004$ | $0.30 \pm 0.005$ | $3.64 \times 10^{-5}$ | 0.43 |
| | 0.9 | $0.90 \pm 0.004$ | $0.10 \pm 0.009$ | $6.44 \times 10^{-5}$ | 0.43 | $0.90 \pm 0.000$ | $0.30 \pm 0.000$ | $5.14 \times 10^{-6}$ | 0.43 |

| | | q (0.5) | | | | q (0.7) | | | |
|---|---|---|---|---|---|---|---|---|---|
| | | $\hat{p}$ | $\hat{q}$ | $\overline{MSE}$ | $\overline{CA}$ | $\hat{p}$ | $\hat{q}$ | $\overline{MSE}$ | $\overline{CA}$ |
| | 0.1 | $0.12 \pm 0.023$ | $0.54 \pm 0.069$ | $3.11 \times 10^{-3}$ | 0.43 | $0.14 \pm 0.058$ | $0.68 \pm 0.051$ | $3.37 \times 10^{-3}$ | 0.43 |
| | 0.3 | $0.29 \pm 0.030$ | $0.51 \pm 0.036$ | $1.05 \times 10^{-3}$ | 0.43 | $0.29 \pm 0.055$ | $0.74 \pm 0.096$ | $5.58 \times 10^{-3}$ | 0.43 |
| $p$ | 0.5 | $0.44 \pm 0.044$ | $0.57 \pm 0.112$ | $9.64 \times 10^{-3}$ | 0.44 | $0.49 \pm 0.016$ | $0.69 \pm 0.024$ | $3.70 \times 10^{-4}$ | 0.43 |
| | 0.7 | $0.68 \pm 0.040$ | $0.55 \pm 0.112$ | $6.67 \times 10^{-3}$ | 0.44 | $0.69 \pm 0.019$ | $0.73 \pm 0.055$ | $1.76 \times 10^{-3}$ | 0.43 |
| | 0.9 | $0.90 \pm 0.006$ | $0.50 \pm 0.000$ | $7.73 \times 10^{-6}$ | 0.43 | $0.88 \pm 0.030$ | $0.74 \pm 0.077$ | $3.10 \times 10^{-3}$ | 0.47 |

| | | q (0.9) | | | |
|---|---|---|---|---|---|
| | | $\hat{p}$ | $\hat{q}$ | $\overline{MSE}$ | $\overline{CA}$ |
| | 0.1 | $0.12 \pm 0.040$ | $0.73 \pm 0.080$ | $1.71 \times 10^{-2}$ | 0.43 |
| | 0.3 | $0.30 \pm 0.010$ | $0.90 \pm 0.010$ | $9.44 \times 10^{-5}$ | 0.43 |
| $p$ | 0.5 | $0.51 \pm 0.020$ | $0.90 \pm 0.070$ | $2.08 \times 10^{-3}$ | 0.43 |
| | 0.7 | $0.68 \pm 0.030$ | $0.92 \pm 0.040$ | $1.18 \times 10^{-3}$ | 0.49 |
| | 0.9 | $0.95 \pm 0.050$ | $0.84 \pm 0.080$ | $7.45 \times 10^{-3}$ | 0.74 |

## C.4    COMPUTATIONAL EFFICIENCY AND SCALABILITY

The proposed framework is highly parallelizable, as agent-level classification tasks are independent, allowing efficient utilization of computational resources. Scalability was evaluated through synthetic experiments on Barabási–Albert (BA) graphs of varying sizes (up to 10,000 nodes, parameter $m = 2$).

Table 8: Robustness Check for Loopy Graph Across Multiple Values of $p$ and $q$

| | | $q$ (0.1) | | | | | $q$ (0.3) | | | |
| --- | --- | --- | --- | --- | --- | --- | --- | --- | --- |
| | | $\hat{p}$ | $\hat{q}$ | $\overline{MSE}$ | $\overline{CA}$ | $\hat{p}$ | $\hat{q}$ | $\overline{MSE}$ | $\overline{CA}$ |
| | 0.1 | $0.10 \pm 0.005$ | $0.10 \pm 0.004$ | $1.98 \times 10^{-5}$ | 0.43 | $0.11 \pm 0.017$ | $0.28 \pm 0.019$ | $4.30 \times 10^{-4}$ | 0.43 |
| $p$ | 0.3 | $0.30 \pm 0.000$ | $0.10 \pm 0.005$ | $1.24 \times 10^{-5}$ | 0.43 | $0.30 \pm 0.009$ | $0.31 \pm 0.005$ | $4.79 \times 10^{-5}$ | 0.43 |
| | 0.5 | $0.50 \pm 0.004$ | $0.10 \pm 0.000$ | $1.23 \times 10^{-5}$ | 0.44 | $0.50 \pm 0.006$ | $0.30 \pm 0.000$ | $1.40 \times 10^{-5}$ | 0.43 |
| | 0.7 | $0.70 \pm 0.005$ | $0.10 \pm 0.000$ | $2.05 \times 10^{-5}$ | 0.43 | $0.70 \pm 0.000$ | $0.30 \pm 0.000$ | $3.45 \times 10^{-6}$ | 0.43 |
| | 0.9 | $0.90 \pm 0.005$ | $0.10 \pm 0.000$ | $3.06 \times 10^{-5}$ | 0.43 | $0.89 \pm 0.014$ | $0.30 \pm 0.000$ | $6.73 \times 10^{-5}$ | 0.43 |

| | | $q$ (0.5) | | | | | $q$ (0.7) | | | |
| --- | --- | --- | --- | --- | --- | --- | --- | --- | --- |
| | | $\hat{p}$ | $\hat{q}$ | $\overline{MSE}$ | $\overline{CA}$ | $\hat{p}$ | $\hat{q}$ | $\overline{MSE}$ | $\overline{CA}$ |
| | 0.1 | $0.09 \pm 0.015$ | $0.53 \pm 0.052$ | $1.78 \times 10^{-3}$ | 0.43 | $0.11 \pm 0.023$ | $0.70 \pm 0.057$ | $1.69 \times 10^{-3}$ | 0.43 |
| | 0.3 | $0.30 \pm 0.005$ | $0.50 \pm 0.017$ | $1.44 \times 10^{-4}$ | 0.48 | $0.30 \pm 0.000$ | $0.70 \pm 0.010$ | $7.91 \times 10^{-5}$ | 0.42 |
| $p$ | 0.5 | $0.50 \pm 0.006$ | $0.50 \pm 0.000$ | $9.17 \times 10^{-6}$ | 0.43 | $0.50 \pm 0.000$ | $0.70 \pm 0.000$ | $4.06 \times 10^{-6}$ | 0.43 |
| | 0.7 | $0.70 \pm 0.000$ | $0.50 \pm 0.000$ | $6.93 \times 10^{-6}$ | 0.43 | $0.70 \pm 0.000$ | $0.70 \pm 0.000$ | $2.37 \times 10^{-6}$ | 0.43 |
| | 0.9 | $0.90 \pm 0.005$ | $0.50 \pm 0.000$ | $1.33 \times 10^{-5}$ | 0.43 | $0.90 \pm 0.007$ | $0.70 \pm 0.000$ | $2.27 \times 10^{-5}$ | 0.43 |

| | | $q$ (0.9) | | | |
| --- | --- | --- | --- | --- | --- |
| | | $\hat{p}$ | $\hat{q}$ | $\overline{MSE}$ | $\overline{CA}$ |
| | 0.1 | $0.11 \pm 0.021$ | $0.84 \pm 0.109$ | $6.74 \times 10^{-3}$ | 0.44 |
| | 0.3 | $0.31 \pm 0.005$ | $0.87 \pm 0.025$ | $9.10 \times 10^{-4}$ | 0.43 |
| $p$ | 0.5 | $0.50 \pm 0.000$ | $0.90 \pm 0.000$ | $2.76 \times 10^{-7}$ | 0.43 |
| | 0.7 | $0.70 \pm 0.000$ | $0.90 \pm 0.000$ | $1.35 \times 10^{-6}$ | 0.43 |
| | 0.9 | $0.91 \pm 0.008$ | $0.90 \pm 0.000$ | $4.50 \times 10^{-5}$ | 0.43 |

Table 9: Robustness Check of Empirical Insiders Graph Across Multiple Values of $p$ and $q$

| | | $q$ (0.1) | | | | | $q$ (0.3) | | | |
| --- | --- | --- | --- | --- | --- | --- | --- | --- | --- |
| | | $\hat{p}$ | $\hat{q}$ | $\overline{MSE}$ | $\overline{CA}$ | $\hat{p}$ | $\hat{q}$ | $\overline{MSE}$ | $\overline{CA}$ |
| | 0.1 | $0.10 \pm 0.009$ | $0.10 \pm 0.005$ | $7.43 \times 10^{-5}$ | $1.62 \times 10^{-2}$ | $0.11 \pm 0.012$ | $0.30 \pm 0.009$ | $1.35 \times 10^{-4}$ | $1.66 \times 10^{-2}$ |
| | 0.3 | $0.37 \pm 0.027$ | $0.10 \pm 0.009$ | $2.66 \times 10^{-3}$ | $1.07 \times 10^{-2}$ | $0.38 \pm 0.041$ | $0.30 \pm 0.010$ | $3.74 \times 10^{-3}$ | $1.16 \times 10^{-2}$ |
| $p$ | 0.5 | $0.54 \pm 0.024$ | $0.10 \pm 0.005$ | $8.67 \times 10^{-4}$ | $6.04 \times 10^{-3}$ | $0.55 \pm 0.059$ | $0.30 \pm 0.007$ | $2.69 \times 10^{-3}$ | $6.96 \times 10^{-3}$ |
| | 0.7 | $0.71 \pm 0.007$ | $0.10 \pm 0.008$ | $1.07 \times 10^{-4}$ | $4.83 \times 10^{-3}$ | $0.72 \pm 0.027$ | $0.30 \pm 0.008$ | $5.98 \times 10^{-4}$ | $6.45 \times 10^{-3}$ |
| | 0.9 | $0.89 \pm 0.022$ | $0.10 \pm 0.005$ | $2.21 \times 10^{-4}$ | $4.75 \times 10^{-3}$ | $0.91 \pm 0.025$ | $0.30 \pm 0.011$ | $3.45 \times 10^{-4}$ | $5.88 \times 10^{-3}$ |

| | | $q$ (0.5) | | | | | $q$ (0.7) | | | |
| --- | --- | --- | --- | --- | --- | --- | --- | --- | --- |
| | | $\hat{p}$ | $\hat{q}$ | $\overline{MSE}$ | $\overline{CA}$ | $\hat{p}$ | $\hat{q}$ | $\overline{MSE}$ | $\overline{CA}$ |
| | 0.1 | $0.11 \pm 0.013$ | $0.49 \pm 0.008$ | $1.16 \times 10^{-4}$ | $1.68 \times 10^{-2}$ | $0.11 \pm 0.004$ | $0.69 \pm 0.016$ | $2.80 \times 10^{-4}$ | $1.67 \times 10^{-2}$ |
| | 0.3 | $0.36 \pm 0.033$ | $0.50 \pm 0.004$ | $2.43 \times 10^{-3}$ | $1.18 \times 10^{-2}$ | $0.37 \pm 0.019$ | $0.69 \pm 0.005$ | $2.53 \times 10^{-3}$ | $1.18 \times 10^{-2}$ |
| $p$ | 0.5 | $0.56 \pm 0.036$ | $0.50 \pm 0.005$ | $2.36 \times 10^{-3}$ | $7.89 \times 10^{-3}$ | $0.53 \pm 0.021$ | $0.70 \pm 0.007$ | $7.28 \times 10^{-4}$ | $7.12 \times 10^{-3}$ |
| | 0.7 | $0.75 \pm 0.019$ | $0.50 \pm 0.011$ | $1.34 \times 10^{-3}$ | $7.08 \times 10^{-3}$ | $0.72 \pm 0.027$ | $0.70 \pm 0.015$ | $5.41 \times 10^{-4}$ | $6.71 \times 10^{-3}$ |
| | 0.9 | $0.91 \pm 0.036$ | $0.50 \pm 0.000$ | $6.06 \times 10^{-4}$ | $6.57 \times 10^{-3}$ | $0.91 \pm 0.046$ | $0.70 \pm 0.005$ | $8.15 \times 10^{-4}$ | $6.34 \times 10^{-3}$ |

| | | $q$ (0.9) | | | |
| --- | --- | --- | --- | --- | --- |
| | | $\hat{p}$ | $\hat{q}$ | $\overline{MSE}$ | $\overline{CA}$ |
| | 0.1 | $0.11 \pm 0.005$ | $0.90 \pm 0.011$ | $7.49 \times 10^{-5}$ | $1.65 \times 10^{-2}$ |
| | 0.3 | $0.33 \pm 0.013$ | $0.90 \pm 0.008$ | $5.44 \times 10^{-4}$ | $1.06 \times 10^{-2}$ |
| p | 0.5 | $0.51 \pm 0.004$ | $0.90 \pm 0.005$ | $6.42 \times 10^{-5}$ | $5.89 \times 10^{-3}$ |
| | 0.7 | $0.72 \pm 0.019$ | $0.91 \pm 0.005$ | $5.01 \times 10^{-4}$ | $5.44 \times 10^{-3}$ |
| | 0.9 | $0.91 \pm 0.022$ | $0.91 \pm 0.005$ | $2.74 \times 10^{-4}$ | $4.70 \times 10^{-3}$ |

As shown in Table 10, DC exhibits sublinear to slightly superlinear growth, with empirical complexity approximately $T(n) \approx k \times n^{0.5-0.65}$. For a 10,000-node BA graph, the average runtime was approximately 1,279 s under parallel execution. By comparison, the GCN baseline required over 6,200 s for a 1,000-node graph and is projected to exceed 122,000 s for 10,000 nodes, making it impractical for large-scale scenarios. These results highlight that DC scales efficiently to large networks while substantially reducing computational cost compared to deep learning–based alternatives.

Table 10: Comparison of execution time (in seconds) between our method DC and the GCN baseline on tree-structured networks, using a MacBook M1 Pro with parallel execution on 10 CPU cores.

| Network Size | DC (Avg $\pm$ Std) | GCN (Avg $\pm$ Std) |
| --- | --- | --- |
| 100 | $89.9 \pm 3.2$ | $169.2 \pm 2.5$ |
| 1000 | $289.6 \pm 11.8$ | $6299.9 \pm 275.0$ |
| 10000 | $1279.0 \pm 47.7$ | $\sim 122{,}000$ (estimated) |

## D ALGORITHMS

Algorithm 1 outlines the process of constructing feature vectors for both synthetic graphs and empirical graphs (when the ground truth is known). The procedure for feature extraction using real ground truth on empirical data is described separately in Algorithm 4. Algorithm 2 and Algorithm 3 detail the optimization process: the former iteratively updates transmission and symptom probabilities until convergence to the underlying ground truth, while the latter fine-tunes the classifier's hyperparameters based on the generated feature vectors.

---

**Algorithm 1** Extract Feature Vector

---

1: **Input:** $\theta$: Observed parameters, $\hat{\theta}$: Initial parameters
2: **Input:** label $\in \{0, 1\}$ indicates observed (0) and simulated(1) data
3: **Output:** $\{\text{feature\_vector}_m\}_{m=1}^{M}$: Set of $M$ labeled feature vectors

4: **function** GENERATEFEATUREVECTOR(label, $\theta$, $\hat{\theta}$)
5:      **for** each feature vector $m = 1$ to $M$ **do**              $\triangleright$ $M$: Number of feature vectors
6:          **for** each simulation $n = 1$ to $N$ **do**              $\triangleright$ $N$: Number of cascades
7:              **if** label $== 1$ **then**
8:                  Simulate spreading cascade with $\hat{\theta}$
9:              **else**
10:                Simulate spreading cascade with $\theta$
11:              **end if**
12:              Extract symptom vector $\mathbf{s}_n \in \mathbb{R}^E$          $\triangleright$ $E$: Number of entities
13:          **end for**
14:          Stack $\{\mathbf{s}_n\}_{n=1}^{N}$ horizontally to form matrix $\mathbf{S}_m \in \mathbb{R}^{E \times N}$
15:          Extract summary statistics $\mathbf{f}_m \in \mathbb{R}^{E \times d}$ from $\mathbf{S}_m$
16:          feature\_vector$_m \leftarrow$ append($\mathbf{f}_m$, label) $\in \mathbb{R}^{E \times (d+1)}$
17:      **end for**
18:      **return** $\{\text{feature\_vector}_m\}_{m=1}^{M}$
19: **end function**

---

---

**Algorithm 2** Learning Optimal Spreading Model's Parameters $\theta^*$

---

1: **Output:** $\hat{\theta}_{\text{pre\_opt}}$: Optimal TP

2: $\mathcal{F}_{\text{GT}} \leftarrow$ GENERATEFEATUREVECTOR(label(0), $\theta$)          $\triangleright$ Ground truth features
3: **while** $\hat{\theta}$ not converged **do**
4:      $\mathcal{F}_{\text{pred}} \leftarrow$ GENERATEFEATUREVECTOR(label(1), $\hat{\theta}$)          $\triangleright$ Predicted features
5:      **for** each entity $v \in \mathcal{V}$ **do**
6:          accuracy$[v] \leftarrow f_{\text{classifier}}^{(v)}(\mathcal{F}_{\text{GT}}[v], \mathcal{F}_{\text{pred}}[v])$          $\triangleright$ Entity-specific classifier
7:      **end for**
8:      $\hat{\theta} \leftarrow$ OPTIMIZER($\hat{\theta}$, $\sum_{v \in \mathcal{V}}$ accuracy$[v]$)
9: **end while**
10: **return** $\hat{\theta}_{\text{pre\_opt}} \leftarrow \hat{\theta}$

---

**Algorithm 3** Entity-Specific Classifier Tuning

1: **Input:** $\mathcal{F}_{\text{GT}}$: Ground truth features, $\mathcal{F}_{\text{pred}}^{opt}$: Predicted features generated using $\hat{\theta}_{\text{pre\_opt}}$

2: **Output:** $\hat{\theta}_{\text{opt}}$: Optimal TP with tuned entity-specific classifiers

3: **for** each entity $v \in \mathcal{V}$ **do**

4:     $\psi_v \leftarrow \text{FINDHYPERPARAM}(f_{\text{classifier}}^{(v)}(\mathcal{F}_{\text{GT}}[v], \mathcal{F}_{\text{pred}}[v]))$     $\triangleright$ Tune classifier hyperparameters

5: **end for**

6: Repeat lines 4–10 of Algorithm 2, replacing each $f_{\text{classifier}}^{(v)}$ with $f_{\text{classifier},\psi_v}^{(v)}$

7: **return** $\hat{\theta}_{\text{opt}} \leftarrow \hat{\theta}$

---

**Algorithm 4** Extract Empirical Feature Vectors

---

1: **Input:** $\mathcal{C} = \{c_1, c_2, \ldots, c_{28}\}$                 ▷ Set of companies
2: **Input:** *is_pre_announcement_mode*          ▷ Flag indicating analysis mode
3: **Output:** $\mathcal{F}_{\text{GT}}^{\text{train}}, \mathcal{F}_{\text{GT}}^{\text{test}}$         ▷ Training and testing feature vectors

4: **function** EXTRACTEMPIRICALGROUNDTRUTHFEATUREVECTORS
5:      Initialize $\mathcal{F}_{\text{GT}}^{\text{train}}, \mathcal{F}_{\text{GT}}^{\text{test}} \leftarrow \emptyset$
6:      **for** each company $c_i \in \mathcal{C}$ **do**
7:          Initialize trade matrix $\mathbf{T}_i \leftarrow \emptyset$
8:          **if** *is_pre_announcement_mode* **then**
9:              $\mathcal{T}_i \leftarrow \mathcal{A}_i, \Delta \leftarrow 1$         ▷ Announcement days and 1-day return
10:          **else**
11:              $\mathcal{T}_i \leftarrow$ non-announcement trade days, $\Delta \leftarrow 5$     ▷ Regular days and 5-day return
12:          **end if**
13:          **for** each trade day $t \in \mathcal{T}_i$ **do**
14:              Initialize vector $\mathbf{v}_t \leftarrow []$         ▷ Vector for all investors at time $t$
15:              **for** each investor $u \in \mathcal{U}_i$ **do**
16:                  **if** *is_pre_announcement_mode* **then**
17:                     $\bar{p}_u \leftarrow$ mean price of trades on the last trade day in the 4-day pre-ann window
18:                  **else**
19:                     $\bar{p}_u \leftarrow$ mean price of $u$'s transactions on $t$
20:                  **end if**
21:                  **if** $p_u$ is defined **then**
22:                     $p_t^{+\Delta} \leftarrow$ market price $\Delta$ days after $t$
23:                     $\mathbf{v}_t \leftarrow \mathbf{v}_t \cup \begin{cases} +1, & \text{if } p_u < p_t^{+\Delta} \\ -1, & \text{otherwise} \end{cases}$     ▷ Profitability encoding
24:                  **else**
25:                     $\mathbf{v}_t \leftarrow \mathbf{v}_t \cup \{0\}$         ▷ No trade found
26:                  **end if**
27:              **end for**
28:              Stack $\mathbf{v}_t$ as new column in $\mathbf{T}_i$
29:          **end for**
30:          Let $n = |\mathcal{T}_i|, n_1 \leftarrow \lfloor 0.6 \cdot n \rfloor$
31:          Initialize $\mathcal{F}_{\text{GT},i}^{\text{train}}, \mathcal{F}_{\text{GT},i}^{\text{test}} \leftarrow \emptyset$
32:          **for** each bootstrap sample $j$ of $m$ columns from $\mathbf{T}_i$ **do**
33:              Extract submatrix $\mathbf{T}_i^{(j)}$
34:              **for** each investor $u$ (row) in $\mathbf{T}_i^{(j)}$ **do**
35:                  Compute summary $\mathbf{f}_u$ (mean, std, positive ratio, etc.)
36:                  **if** $j$ is from first $n_1$ columns **then**
37:                     Add $(\mathbf{f}_u, \text{label}(0))$ to $\mathcal{F}_{\text{GT},i}^{\text{train}}$
38:                  **else**
39:                     Add $(\mathbf{f}_u, \text{label}(0))$ to $\mathcal{F}_{\text{GT},i}^{\text{test}}$
40:                  **end if**
41:              **end for**
42:          **end for**
43:          $\mathcal{F}_{\text{GT}}^{\text{train}} \leftarrow \mathcal{F}_{\text{GT}}^{\text{train}} \cup \mathcal{F}_{\text{GT},i}^{\text{train}}$
44:          $\mathcal{F}_{\text{GT}}^{\text{test}} \leftarrow \mathcal{F}_{\text{GT}}^{\text{test}} \cup \mathcal{F}_{\text{GT},i}^{\text{test}}$
45:      **end for**
46:      **return** $\mathcal{F}_{\text{GT}}^{\text{train}}, \mathcal{F}_{\text{GT}}^{\text{test}}$
47: **end function**

---

# E  HYPERPARAMETER SEARCH SPACE

We conduct a grid search over the following hyperparameter spaces for each classifier:

- **Support Vector Machine (SVM)**: $C \in \{1, 10, 100, 1000\}$, kernel $\in \{\text{linear}, \text{rbf}, \text{poly}\}$, $\gamma \in \{0.01, 0.1, 1\}$
- **Random Forest (RF)**: $n\_estimators \in \{100, 200\}$, max_depth $\in \{\text{None}, 10, 20\}$, min_samples_split $\in \{2, 5\}$
- **Naive Bayes (NB)**: var_smoothing $\in \{10^{-9}, 10^{-8}, 10^{-7}, 10^{-6}\}$
- **Stochastic Gradient Descent (SGD)**: loss $\in \{\text{hinge}, \text{log}\}$, penalty $\in \{\ell_2, \text{elasticnet}\}$, $\alpha \in \{10^{-4}, 10^{-3}\}$
- **Decision Tree (DT)**: criterion $\in \{\text{gini}, \text{entropy}\}$, max_depth $\in \{\text{None}, 10, 20\}$, min_samples_split $\in \{2, 5\}$
- $k$**-Nearest Neighbors (KNN)**: $n\_neighbors \in \{5, 10\}$, weights $\in \{\text{uniform}, \text{distance}\}$, algorithm $\in \{\text{auto}, \text{ball\_tree}\}$
- **Logistic Regression (LR)**: penalty $\in \{\ell_2\}$, $C \in \{0.01, 0.1, 1\}$, solver $\in \{\text{liblinear}\}$, max_iter $\in \{100, 200\}$

# F  NETWORK STRUCTURES

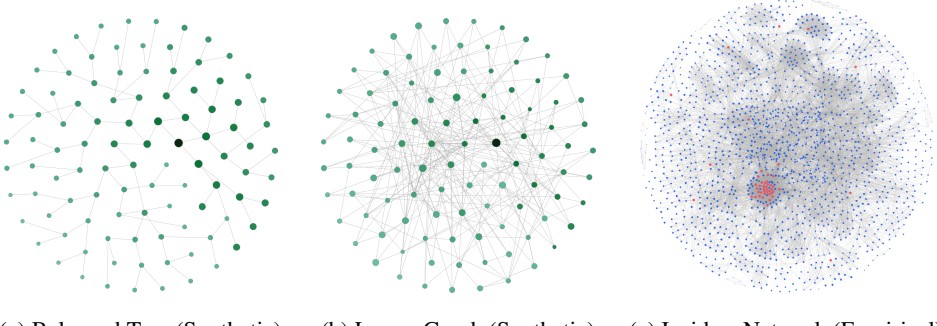

(a) Balanced Tree (Synthetic)     (b) Loopy Graph (Synthetic)     (c) Insiders Network (Empirical)

Figure 3: Network topologies used for evaluating the proposed framework. (a) A balanced tree graph with 198 edges, used to simulate a hierarchical spread process. (b) A loopy synthetic graph with 398 edges, capturing richer connectivity and feedback loops. (c) An empirical network derived from insider trading data, comprising 32,925 edges and 1,661 investor nodes. In (a) and (b), a single seed node is marked in dark green; node size and color reflect distance from the seed (closer nodes appear larger and darker). In (c), multiple seed nodes are present and highlighted in red.

# G  DESCRIPTIVE STATISTICS AND RESULTS ON EMPIRICAL ANALYSIS

Table 11 provides a summary of investor transaction activity across all companies. Panel A presents aggregate company-level statistics, while Panels B and C focus on transactions during the pre-announcement and non-announcement periods, respectively.

Table 12 presents the number of investors *(Inv)* and transaction records associated with each company. It includes the number of transactions (*Trans*) during the pre-announcement period (*Pre-Ann*), the non-announcement period (*Non-Ann*), and the total number of transactions. Transactions occurring on the exact day of the announcement are excluded from both periods, which means the sum of *Pre-Ann* and *Non-Ann* transactions may be less than the total count. The *average baseline trade probabilities* of all investors associated with each company are presented in Table 13. These baseline trading behaviors reflect investor activity outside the pre-announcement period. Table 14 presents the inferred $\hat{p}$ and $\hat{q}$ during the announcement and non-announcement periods, along with their corresponding ratios.

Table 11: Descriptive statistic of Insiders data

| | Mean | Min | Q1 | Median | Q3 | Max | Standard Deviation |
|---|---|---|---|---|---|---|---|
| *Panel: A* Complete dataset | | | | | | | |
| Number of investors | 298.96 | 115 | 219.25 | 282.5 | 366.5 | 595 | 112.96 |
| Number of board members | 14.5 | 4 | 9.75 | 12.5 | 18.25 | 46 | 8.56 |
| Number of trades | 5130.57 | 899 | 2689 | 4730.5 | 6220.5 | 12548 | 2953.98 |
| Number of announcements | 233.36 | 90 | 165.5 | 197 | 288 | 526 | 109.99 |
| *Panel: B* Pre-Announcement Period (Pre-Ann) | | | | | | | |
| Investors in Pre-Announcement Period | 179.57 | 57 | 140 | 163.5 | 236 | 345 | 69.88 |
| Trades in Pre-Announcement Period | 1582.61 | 289 | 932.25 | 1626.5 | 1975.5 | 3125 | 812.50 |
| *Panel: C* Non-Announcement Period (Non-Ann) | | | | | | | |
| Investors in Non-Announcement Period | 234.96 | 82 | 155.5 | 230.5 | 284.5 | 497 | 101.48 |
| Trades in Non-Announcement Period | 2809.71 | 489 | 1398 | 2296 | 3925.75 | 8056 | 1866.03 |

Table 12: Descriptive statistics with each company

| Company | Full period | | | | Pre-Ann | | Non-Ann | |
|---|---|---|---|---|---|---|---|---|
| | Inv | Seed | Trans | Ann | Trans | Inv | Trans | Inv |
| Amer Sports | 198 | 18 | 2742 | 291 | 973 | 119 | 1415 | 134 |
| Cargotec | 265 | 15 | 12548 | 244 | 2948 | 175 | 8056 | 180 |
| Comptel | 115 | 10 | 899 | 121 | 299 | 68 | 489 | 82 |
| Elisa | 284 | 20 | 2465 | 189 | 816 | 163 | 1347 | 206 |
| F-Secure | 124 | 8 | 1245 | 95 | 289 | 57 | 839 | 98 |
| Fortum | 398 | 18 | 4921 | 184 | 1888 | 258 | 2274 | 283 |
| Huhtamäki | 234 | 10 | 3443 | 159 | 1023 | 143 | 1868 | 196 |
| Kemira | 312 | 19 | 6120 | 187 | 1499 | 157 | 3979 | 289 |
| Kesko | 243 | 5 | 4033 | 413 | 1663 | 164 | 1642 | 180 |
| Kone | 281 | 13 | 9767 | 167 | 3047 | 145 | 5817 | 225 |
| Konecranes | 196 | 17 | 2511 | 319 | 912 | 104 | 1157 | 154 |
| Metsa | 223 | 11 | 4292 | 254 | 1632 | 143 | 2088 | 167 |
| Metso | 360 | 7 | 6104 | 419 | 2484 | 247 | 2369 | 277 |
| Neste | 409 | 19 | 5285 | 161 | 1536 | 230 | 3236 | 337 |
| Nokia | 595 | 46 | 9994 | 180 | 2238 | 345 | 6084 | 497 |
| Nokian Renkaat | 314 | 10 | 4123 | 93 | 977 | 187 | 2318 | 262 |
| Nordea | 463 | 24 | 5529 | 198 | 1717 | 268 | 3211 | 403 |
| Outotec | 176 | 8 | 2530 | 122 | 723 | 87 | 1467 | 149 |
| Rautaruukki | 320 | 12 | 4540 | 287 | 1621 | 239 | 2077 | 239 |
| Sampo | 458 | 26 | 6522 | 287 | 1649 | 250 | 3908 | 403 |
| Sanoma | 308 | 4 | 7250 | 213 | 2259 | 187 | 4403 | 238 |
| Stockmann | 217 | 4 | 2197 | 169 | 885 | 131 | 951 | 156 |
| Stora Enso | 197 | 23 | 1826 | 386 | 791 | 154 | 675 | 110 |
| Tieto | 220 | 10 | 2785 | 526 | 939 | 150 | 1226 | 134 |
| UPM-Kymmene | 473 | 15 | 8962 | 196 | 3125 | 300 | 4233 | 354 |
| Uponor | 272 | 13 | 5952 | 90 | 1776 | 127 | 3448 | 236 |
| Wärtsilä | 330 | 12 | 9697 | 366 | 2955 | 195 | 5339 | 264 |
| YIT | 386 | 9 | 5374 | 218 | 1649 | 235 | 2756 | 326 |

Table 13: Baseline trade probabilities by company

| Company | $\bar{b}_1$ | $\bar{b}_2$ | $\bar{b}_0$ |
|---|---|---|---|
| Company 1 | $3.35 \times 10^{-3}$ | $3.65 \times 10^{-3}$ | $9.93 \times 10^{-1}$ |
| Company 2 | $5.22 \times 10^{-3}$ | $5.13 \times 10^{-3}$ | $9.90 \times 10^{-1}$ |
| Company 3 | $4.26 \times 10^{-3}$ | $4.91 \times 10^{-3}$ | $9.91 \times 10^{-1}$ |
| Company 4 | $3.93 \times 10^{-3}$ | $3.34 \times 10^{-3}$ | $9.93 \times 10^{-1}$ |
| Company 5 | $4.59 \times 10^{-3}$ | $4.88 \times 10^{-3}$ | $9.90 \times 10^{-1}$ |
| Company 6 | $4.18 \times 10^{-3}$ | $4.37 \times 10^{-3}$ | $9.91 \times 10^{-1}$ |
| Company 7 | $6.77 \times 10^{-3}$ | $6.08 \times 10^{-3}$ | $9.87 \times 10^{-1}$ |
| Company 8 | $4.68 \times 10^{-3}$ | $4.84 \times 10^{-3}$ | $9.90 \times 10^{-1}$ |
| Company 9 | $4.77 \times 10^{-3}$ | $5.16 \times 10^{-3}$ | $9.90 \times 10^{-1}$ |
| Company 10 | $3.84 \times 10^{-3}$ | $4.22 \times 10^{-3}$ | $9.92 \times 10^{-1}$ |
| Company 11 | $3.67 \times 10^{-3}$ | $4.22 \times 10^{-3}$ | $9.92 \times 10^{-1}$ |
| Company 12 | $3.42 \times 10^{-3}$ | $3.22 \times 10^{-3}$ | $9.93 \times 10^{-1}$ |
| Company 13 | $5.31 \times 10^{-3}$ | $7.07 \times 10^{-3}$ | $9.88 \times 10^{-1}$ |
| Company 14 | $5.11 \times 10^{-3}$ | $3.75 \times 10^{-3}$ | $9.91 \times 10^{-1}$ |
| Company 15 | $3.75 \times 10^{-3}$ | $3.80 \times 10^{-3}$ | $9.92 \times 10^{-1}$ |
| Company 16 | $2.96 \times 10^{-3}$ | $3.89 \times 10^{-3}$ | $9.93 \times 10^{-1}$ |
| Company 17 | $3.22 \times 10^{-3}$ | $3.69 \times 10^{-3}$ | $9.93 \times 10^{-1}$ |
| Company 18 | $4.02 \times 10^{-3}$ | $4.57 \times 10^{-3}$ | $9.91 \times 10^{-1}$ |
| Company 19 | $3.97 \times 10^{-3}$ | $5.97 \times 10^{-3}$ | $9.90 \times 10^{-1}$ |
| Company 20 | $4.46 \times 10^{-3}$ | $5.79 \times 10^{-3}$ | $9.90 \times 10^{-1}$ |
| Company 21 | $3.99 \times 10^{-3}$ | $4.50 \times 10^{-3}$ | $9.91 \times 10^{-1}$ |
| Company 22 | $5.09 \times 10^{-3}$ | $5.48 \times 10^{-3}$ | $9.89 \times 10^{-1}$ |
| Company 23 | $6.02 \times 10^{-3}$ | $5.46 \times 10^{-3}$ | $9.88 \times 10^{-1}$ |
| Company 24 | $4.85 \times 10^{-3}$ | $4.31 \times 10^{-3}$ | $9.90 \times 10^{-1}$ |
| Company 25 | $4.28 \times 10^{-3}$ | $3.78 \times 10^{-3}$ | $9.92 \times 10^{-1}$ |
| Company 26 | $3.70 \times 10^{-3}$ | $3.49 \times 10^{-3}$ | $9.93 \times 10^{-1}$ |
| Company 27 | $3.50 \times 10^{-3}$ | $3.17 \times 10^{-3}$ | $9.93 \times 10^{-1}$ |
| Company 28 | $7.24 \times 10^{-3}$ | $8.62 \times 10^{-3}$ | $9.84 \times 10^{-1}$ |

Table 14: Inferred transmission probablities of Insiders network

| Company | Pre-Announcement Period | | | Non-Announcement Period | | | Ratio | |
|---|---|---|---|---|---|---|---|---|
| | $\overline{\hat{p}_a}$ | $\overline{\hat{q}_a}$ | $\overline{CA_a}$ | $\overline{\hat{p}_n}$ | $\overline{\hat{q}_n}$ | $\overline{CA_n}$ | $\overline{\hat{p}_a}/\overline{\hat{p}_n}$ | $\overline{\hat{q}_a}/\overline{\hat{q}_n}$ |
| Company 1 | 0.39 | 0.12 | $2.22 \times 10^{-2}\%$ | 0.18 | 0.02 | $1.19 \times 10^{-2}\%$ | 2.09 | 6.30 |
| Company 2 | 0.59 | 0.03 | $2.93 \times 10^{-2}\%$ | 0.30 | 0.01 | $1.41 \times 10^{-2}\%$ | 1.97 | 2.07 |
| Company 3 | 0.60 | 0.04 | $2.57 \times 10^{-2}\%$ | 0.32 | 0.02 | $1.89 \times 10^{-2}\%$ | 1.86 | 1.69 |
| Company 4 | 0.53 | 0.03 | $3.30 \times 10^{-2}\%$ | 0.29 | 0.06 | $1.41 \times 10^{-2}\%$ | 1.81 | 0.53 |
| Company 5 | 0.40 | 0.06 | $1.85 \times 10^{-2}\%$ | 0.23 | 0.05 | $1.21 \times 10^{-2}\%$ | 1.74 | 1.08 |
| Company 6 | 0.50 | 0.03 | $2.04 \times 10^{-2}\%$ | 0.31 | 0.02 | $1.72 \times 10^{-2}\%$ | 1.63 | 1.60 |
| Company 7 | 0.57 | 0.03 | $2.21 \times 10^{-2}\%$ | 0.36 | 0.04 | $2.58 \times 10^{-2}\%$ | 1.61 | 0.66 |
| Company 8 | 0.52 | 0.03 | $2.64 \times 10^{-2}\%$ | 0.35 | 0.06 | $2.67 \times 10^{-2}\%$ | 1.48 | 0.45 |
| Company 9 | 0.57 | 0.02 | $1.96 \times 10^{-2}\%$ | 0.38 | 0.03 | $2.32 \times 10^{-2}\%$ | 1.47 | 0.73 |
| Company 10 | 0.47 | 0.04 | $2.71 \times 10^{-2}\%$ | 0.36 | 0.03 | $1.64 \times 10^{-2}\%$ | 1.30 | 1.14 |
| Company 11 | 0.35 | 0.09 | $2.00 \times 10^{-2}\%$ | 0.28 | 0.05 | $1.52 \times 10^{-2}\%$ | 1.26 | 1.79 |
| Company 12 | 0.36 | 0.03 | $1.94 \times 10^{-2}\%$ | 0.29 | 0.01 | $1.76 \times 10^{-2}\%$ | 1.23 | 2.08 |
| Company 13 | 0.58 | 0.05 | $3.56 \times 10^{-2}\%$ | 0.48 | 0.02 | $3.15 \times 10^{-2}\%$ | 1.22 | 2.63 |
| Company 14 | 0.55 | 0.04 | $1.91 \times 10^{-2}\%$ | 0.46 | 0.02 | $1.59 \times 10^{-2}\%$ | 1.22 | 2.57 |
| Company 15 | 0.49 | 0.02 | $1.66 \times 10^{-2}\%$ | 0.41 | 0.02 | $2.20 \times 10^{-2}\%$ | 1.19 | 1.37 |
| Company 16 | 0.35 | 0.06 | $2.20 \times 10^{-2}\%$ | 0.31 | 0.04 | $1.48 \times 10^{-2}\%$ | 1.13 | 1.40 |
| Company 17 | 0.36 | 0.04 | $2.75 \times 10^{-2}\%$ | 0.33 | 0.01 | $1.28 \times 10^{-2}\%$ | 1.07 | 2.73 |
| Company 18 | 0.42 | 0.05 | $9.61 \times 10^{-3}\%$ | 0.40 | 0.06 | $2.30 \times 10^{-2}\%$ | 1.07 | 0.76 |
| Company 19 | 0.36 | 0.07 | $2.33 \times 10^{-2}\%$ | 0.35 | 0.03 | $2.07 \times 10^{-2}\%$ | 1.04 | 2.62 |
| Company 20 | 0.48 | 0.03 | $2.44 \times 10^{-2}\%$ | 0.48 | 0.02 | $2.49 \times 10^{-2}\%$ | 1.01 | 1.69 |
| Company 21 | 0.41 | 0.03 | $1.87 \times 10^{-2}\%$ | 0.41 | 0.01 | $1.57 \times 10^{-2}\%$ | 1.00 | 2.32 |
| Company 22 | 0.44 | 0.04 | $3.49 \times 10^{-2}\%$ | 0.45 | 0.01 | $1.85 \times 10^{-2}\%$ | 0.99 | 2.80 |
| Company 23 | 0.42 | 0.06 | $4.04 \times 10^{-2}\%$ | 0.44 | 0.01 | $1.92 \times 10^{-2}\%$ | 0.95 | 4.06 |
| Company 24 | 0.42 | 0.03 | $1.66 \times 10^{-2}\%$ | 0.45 | 0.05 | $2.01 \times 10^{-2}\%$ | 0.94 | 0.55 |
| Company 25 | 0.41 | 0.03 | $2.64 \times 10^{-2}\%$ | 0.44 | 0.01 | $1.86 \times 10^{-2}\%$ | 0.93 | 2.34 |
| Company 26 | 0.34 | 0.04 | $2.38 \times 10^{-2}\%$ | 0.38 | 0.01 | $1.71 \times 10^{-2}\%$ | 0.89 | 3.04 |
| Company 27 | 0.45 | 0.09 | $2.48 \times 10^{-2}\%$ | 0.66 | 0.01 | $1.76 \times 10^{-2}\%$ | 0.68 | 7.90 |
| Company 28 | 0.34 | 0.04 | $3.34 \times 10^{-2}\%$ | 0.52 | 0.03 | $4.69 \times 10^{-2}\%$ | 0.65 | 1.17 |

