# OpenReview forum: "Learning hidden cascades via classification"
_ICLR.cc/2026/Conference — Submitted to ICLR 2026_

### Official Review · Reviewer_iTa8 · 2025-10-24

**Soundness:** 2
**Presentation:** 2
**Contribution:** 2
**Rating:** 4
**Confidence:** 3

**Summary:**

This paper addresses the challenge of inferring spreading dynamics in networks when node infection statuses are unobservable. Specifically, the authors introduce the Hidden Cascade (HC) problem and propose a Distribution Classification (DC) framework to address it. The method is validated on both extensive synthetic networks and a real-world insider trading network. When compared with Approximate Bayesian Computation (ABC) and several GNN baselines, the DC framework demonstrates superior performance across diverse network types and noise levels.

**Strengths:**

1. The authors introduce a novel problem, termed the Hidden Cascade problem, in which infection states are latent and only indirect symptoms are observable.

2. The proposed two-sided observation model extends the conventional one-sided assumption.

3. The Distribution Classification (DC) framework reformulates MLE-based inference as a classification task that minimizes average classification accuracy, inspired by adversarial distribution matching.

4. The proposed approach is model-agnostic.

**Weaknesses:**

1. The figures (e.g., Figure 1) could be more intuitive and visually informative.

2. This manuscript lacks an ablation study to examine how different feature vectors affect inference performance.

3. The proposed HC problem is conceptually similar to micro-level prediction in diffusion modeling, where the goal is to infer the infection status of each node in a network.

4. More competitive baselines, such as point-process-based models, should be included for comparison.

5. Since the authors claim that their model is also applicable to disease and social domains, real-world validations in these areas would strengthen the paper.

6. The intuitive connection between minimizing classification accuracy and recovering the true diffusion lacks formal justification.

**Questions:**

See weaknesses.

---

### Official Review · Reviewer_P1Wi · 2025-10-30

**Soundness:** 1
**Presentation:** 1
**Contribution:** 1
**Rating:** 0
**Confidence:** 5

**Summary:**

The paper "Learning hidden cascades via classification" proposes a method for learning the parameters of a propagation model on graphs. Specifically, considering a diffusion model such as the Independent Cascade model, the aim is to extract the transmission parameter of an item from one node to its direct neighbors, along with the probability of phenotyps given the state (infected or not) of a node. The setting differs from most of existing models where the state of nodes is observable at the end of the cascade, not only markers of that state. However many classical statistical frameworks could be applied to this simple model. Experiments give some performance results, but the proposed approach is the only experimented (with different kinds of classifiers).

**Strengths:**

NA

**Weaknesses:**

- Very simple model for which many parameter identification approaches could be applied. The authors dismiss models like Markov Random Fields with inference via loopy belief propagation, or Bayesian methods such as Gibbs sampling or EM, claiming they do not scale. I suspect that the proposed method does not scale either, while offering no guarantees against oscillations or divergence. Numerous modern methods based on probabilistic neural networks could equally be applied in this setting.

Positioning and Innovation: The paper claims that few approaches address diffusion processes with missing information. This is far from true. First, many works (e.g., see works from Saito) focus on identifying diffusion parameters without observing transmission events (i.e., who passed information to whom). Closer to the considered setting, other works make assumptions based on probabilistic priors about whether an uninfected node was observed or not. There is a fairly large literature on this topic. Furthermore, the paper ignores the temporal aspect of diffusion models, which is central to them. Without considering temporality, the problem reduces to classical collective inference on graphs, for which many methods already exist. Finally, very little is discussed regarding learning methods related to the proposed approach (see the Innovation section below).

- Innovation: Regarding the proposed method, the authors describe a generation-and-discrimination process and present it as an original contribution. I want to note that this is exactly the framework of, for example, Generative Adversarial Networks (GANs), which the authors should at least discuss.

- Method Viability: The framework is therefore similar to GANs, but unlike standard GANs, the current setting deals with discrete events. In continuous GANs, it is possible to backpropagate discriminator gradients through the generator to guide learning. In the discrete context, this is more complicated, but numerous works using reinforcement learning exist to favor rollouts that are least distinguishable from true observations. The proposed method, which is therefore not new, alternates generator and discriminator optimizations completely independently, which virtually guarantees oscillations or divergence (and classical GANs are already hard to stabilize!). Moreover, the method trains a separate classifier for each node in the graph, which clearly does not scale and carries a high risk of overfitting. The optimization methods used are also quite outdated.

- Presentation: The writing style does not meet the standard of ICLR publications. The paper lacks rigor and formalization in many places. The analysis is brief, and the discussions are insufficiently developed. The algorithms do not help comprehension (for example, there is a detailed algorithm for computing an average, but no global view of how the optimizations are sequenced).

- Experimental Methodology: The experiments do not report any concurrent training approaches. The only reported results compare the use of various classical ML classifiers in the proposed learning scheme, which is clearly not enough.

**Questions:**

- The overall quality metric that the generator is supposed to optimize is not described. Is it an average of the prediction quality over the nodes, as judged by the discriminator?

- Training looks to be performed on fully annotated graphs, as formalization do not mention the possbility of having no label for some nodes in the graph. I understood that we were in a transductive setting, where supervision on some nodes can be transfered on others. Is it not the case?

---

### Official Review · Reviewer_bih2 · 2025-11-03

**Soundness:** 1
**Presentation:** 1
**Contribution:** 2
**Rating:** 6
**Confidence:** 3

**Summary:**

This paper tackles the Hidden Cascade (HC) problem—estimating spreading parameters in networks when only indirect, noisy, symptom-based observations are available. The proposed Distribution Classification (DC) framework formulates parameter inference as a distribution matching task between real and simulated symptom statistics to recover propagation (p) and symptom (q) parameters. Extensive experiments on synthetic and real-world datasets, including an insider trading network, demonstrate that DC outperforms ABC and several GNN baselines in both accuracy and scalability.

**Strengths:**

S1. The paper offers a rigorous and practically meaningful formulation of the HC problem, extending cascade inference to partially observable and noisy scenarios. The motivation, particularly in epidemiological and financial contexts, is compelling.

S2. The key innovation lies in reframing parameter recovery as a distributional classification problem, enabling flexible and fine-grained inference through entity-specific classifiers rather than global aggregation.

S3. The experimental evaluation is thorough, featuring ablations on classifier and statistic design, robustness tests under diverse graph structures, and consistent gains across multiple baselines.

**Weaknesses:**

W1. The paper lacks theoretical or empirical analysis of identifiability for p and q. Without conditions ensuring uniqueness or consistency, the interpretability of the inferred parameters remains uncertain.

W2. Sections 3.1–3.2 could be more rigorous and transparent. Key details such as negative sampling, class imbalance handling, and aggregation of classifier outputs are under-specified, and notation inconsistencies (e.g., q, b_1, b_2) may hinder reproducibility.

W3. Baseline coverage is limited. The exclusion of stronger neural or attention-based cascade models (e.g., DeepCas) weakens the “state-of-the-art” claim. Moreover, hyperparameter tuning based solely on Powell’s convergence warrants further justification.

**Questions:**

Q1. What formal or empirical criteria ensure identifiability of p and q? Could different parameter sets produce indistinguishable statistics?

Q2. How robust is DC to class imbalance or misspecified symptom models, especially under nontrivial background noise?

Q3. How does DC compare to more advanced attention-based or sequence-structural models? If excluded, how does this affect the generality of the claims?

Q4. In the insider trading analysis, how were firm-specific heterogeneities handled? Were placebo or randomization tests used to control for spurious correlations?

---

### Meta-Review · Area_Chair_tDXw · 2025-12-15

**Summary:**

The paper proposes a model for transmission dynamics with latent node factors. They propose an optimization approach that estimates discrepancy between the real and simulated data.
The reviewers had several valid concerns, including a large body of related work not properly discussed, missing baseline in evaluation, lack of more real-world applications (the inside-network analysis is somewhat anecdotal and is not a rigorous comparison), and lack of rigor.
One particularly glaring omission is to GANs, which also employ the idea of maximizing indistinguishability between real and simulated data.

**Reviewer Concerns:**

No rebuttal

**Reviewer Scores:**

No rebuttal

---

### Decision · Program_Chairs · 2026-01-26

Reject